

# Estimation of Canada's methane emissions: inverse modelling analysis using the ECCC measurement network

Misa Ishizawa, Douglas Chan, Doug Worthy, Elton Chan, Felix Vogel, Joe R. Melton, and Vivek K. Arora

5    Climate Research Division, Environment and Climate Change Canada, Canada

*Correspondence to:* Misa Ishizawa (misa.ishizawa@ec.gc.ca) and Douglas Chan (douglas.chan@ec.gc.ca)

**Abstract.** Canada has major sources of atmospheric methane ($CH_4$), with the world second-largest boreal wetland and the world fourth-largest natural gas production. However, Canada's $CH_4$ emissions remain uncertain among estimates. Better quantification and characterization of Canada's $CH_4$ emissions are critical for climate mitigation strategies.  To improve our understanding of Canada's $CH_4$ emissions, we performed an ensemble regional inversion (2007–2017) constrained with the Environment and Climate Change Canada (ECCC) surface measurement network. The decadal $CH_4$ estimates show no significant trend, unlike some studies that reported long-term trends. The total $CH_4$ estimate is 17.4 (15.3–19.5) Tg $CH_4$ year$^{-1}$, partitioned into natural and anthropogenic sources, 10.8 (7.5–13.2) and 6.6 (6.2–7.8) Tg $CH_4$ year$^{-1}$, respectively. The estimated anthropogenic emission is higher than inventories, mainly in western Canada (with the fossil fuel industry). Furthermore, the results reveal notable spatiotemporal characteristics.  First, the modelled gradients of atmospheric $CH_4$ show improvement after inversion when compared to observations, implying the $CH_4$ gradients could help verify the inversion results. Second, the seasonal variations show slow onset and late summer maximum, indicating wetland $CH_4$ flux has hysteretic dependence on air temperature. Third, the boreal winter natural $CH_4$ emissions, usually treated as negligible, appear quantifiable ($\geq 20$ % of annual emissions). Understanding winter emission is important for climate prediction, as the winter in Canada is warming faster than the summer. Fourth, the inter-annual variability in estimated $CH_4$ emissions is positively correlated with summer air temperature anomalies. This could enhance Canada's natural $CH_4$ emission in the warming climate.

## 1 Introduction

Atmospheric methane ($CH_4$) is the second most important long-lived greenhouse gas (GHG) in terms of radiative forcing, contributing about 17 % globally (e.g., Butler and Montzka, 2019). The global atmospheric $CH_4$ level has increased by ~260 % compared to the pre-industrial level (WMO, 2020). Such drastic long-term increase in atmospheric $CH_4$ is mainly attributed to the increasing $CH_4$ emissions through human activities, such as livestock farming, rice cultivation, fossil fuel exploitation and waste disposal (Saunois et al., 2020). Because of the strong radiative forcing and its relatively short lifetime of less than a decade in the atmosphere, there is a global effort to reduce anthropogenic $CH_4$ emissions by at least 30 percent from 2020 levels by 2030 for global climate mitigation (CCAC, 2023). Besides the anthropogenic sources, ~40 % of global $CH_4$ emissions





come from various natural sources. Among them, wetlands are the largest global source of atmospheric $CH_4$, and are likely to
increase under the warming climate (IPCC, 2022). However, due to limited observational constraints and verifications, natural
$CH_4$ emissions are highly uncertain. Top-down estimates constrain the $CH_4$ emissions with atmospheric $CH_4$ observations,
while bottom-up estimates are based on process-based ecosystem models and emission inventory statistics. These different
approaches yield a wide range of $CH_4$ emission estimates. For example, Saunois et al. (2020) reported the average global

emission of 737 Tg $CH_4$ year$^{-1}$ (range of min–max, 594–881) from bottom-up estimates for 2008–2017, which is ~30 % larger
than the top-down results of 596 Tg $CH_4$ year$^{-1}$ (550–594). Regionally, the emission uncertainty could be much larger
(Kirschke et al., 2013; Stavert et al., 2021). Therefore, accurate estimates of $CH_4$ emissions are essential for methane reduction
strategies and future climate mitigation.

Canada has both natural and anthropogenic $CH_4$ emissions. Estimates of Canada's $CH_4$ emissions range widely. The most

significant discrepancy among emission estimates is in the natural $CH_4$ flux estimates. For example, terrestrial ecosystem
models estimate the wetland $CH_4$ emissions from ~10 to 50 Tg $CH_4$ year$^{-1}$ (Poulter et al., 2017). Natural $CH_4$ fluxes are
biogenic fluxes from wetlands, including inland waters, such as lakes, ponds, and rivers, distributing widely and covering ~13
% of Canada's land surface (ECCC, 2016). Furthermore, Canada's natural $CH_4$ source is potentially enhanced by the warming
climate. All the ecosystem models that participated in the latest GCP $CH_4$ project (GCP, 2020) showed positive trends of

wetland $CH_4$ corresponding to increasing air temperature and wetland area (Poulter et al., 2017; Stavert et al., 2021). Top-
down studies reported mixed results for Canada's natural $CH_4$ emission trend. For example, Thompson et al. (2017); Sheng et
al. (2018) found increasing trends, while the ensemble mean of 11 global surface inverse models in the latest GCP $CH_4$ project
(GCP, 2020) showed a gradual downward trend of Canada's wetland $CH_4$ emission over the last two decades, ~-0.3 Tg $CH_4$
year$^{-2}$ (Stavert et al., 2021) and Wittig et al. (2023) reported a slight decreasing trend of wetland emission, -1.4 % year$^{-1}$ for

North America (Canada and Alaska). Thus, the trend of Canada's wetland $CH_4$ emission remains uncertain among various
estimation approaches.

Canada is the fourth largest producer of natural gas, 5 % of world production (https://www.nrcan.gc.ca/science-data/data-
analysis/energy-data-analysis/energy-facts/natural-gas-facts/20067, last access 9 September 2023). According to Canada's
national greenhouse gas inventory report (NIR) (ECCC, 2022), the anthropogenic $CH_4$ emission estimate is on average ~4 Tg

$CH_4$ year$^{-1}$, from oil and gas operations (38 %), agriculture (30 %), waste treatments (28 %) and others (transportation and coal
mining, 4 %). Most fossil fuel $CH_4$ resources are located in the western provinces, British Columbia, Alberta, and
Saskatchewan. Among them, Alberta represents ~63 % of Canada's natural gas production. The NIR reports the western
provinces emit ~70 % of Canada's anthropogenic $CH_4$ emission, but recent studies have estimated much more anthropogenic
$CH_4$ emission than the NIR, especially from oil and gas sectors. Some studies are based on campaign measurements around

the oil and gas production areas (e.g., Baray et al., 2018; Johnson et al., 2017; Johnson et al., 2023), and others are from
modelling studies with observational constraints (e.g., Miller et al., 2014; Thompson et al., 2017; Chan et al., 2020). The



discrepancies in anthropogenic $CH_4$ emission estimates need to be minimized to regulate Canada's anthropogenic $CH_4$ emissions. Observation-based emission estimate would be an important tool to assess if Canada has met the reduction target set in the 2015 Paris Agreement of the United Nations Framework Convention on Climate Change (UNEP, 2021).

Environment and Climate Change Canada (ECCC) has been expanding the GHG monitoring program over the past decades across Canada. These observations have been used in regional inversion studies to estimate $CH_4$ emissions in the Canadian Arctic region (Ishizawa et al. 2019), and anthropogenic $CH_4$ emissions in western Canada (Chan et al. 2020). This regional inversion study focuses on using these ECCC observations to estimate $CH_4$ emissions in Canada. The inverse model used an ensemble of multiple atmospheric transport models and prior fluxes, allowing for the investigation of the sensitivity and
robustness of the estimated fluxes to inversion setups. Section 2 describes the atmospheric measurements, the inverse model and the method for partitioning the total $CH_4$ fluxes into natural and anthropogenic sources. Section 3 presents the results of the inverse model, and discusses the spatiotemporal characteristics of the fluxes and their relationship to climate forcings and the evaluation of the fluxes using the independent flux information contained in the gradient of the observed mixing ratios. The final summary is presented in Sect. 4.

## 75  2 Methods

This section provides a brief description of the atmospheric $CH_4$ data in Canada and the regional inverse model.

### 2.1 Atmospheric $CH_4$ measurements

This study utilized the records of continuous atmospheric $CH_4$ measurements by ECCC's GHG monitoring program across Canada. Among the ECCC-operated sites, we focused on 13 sites (Fig. 1) to constrain Canada's national and subregional $CH_4$
fluxes for 11 years from 2007 to 2017. The chosen sites have at least 5-year records. The location information of the sites used in this study is in Table 1. Brief descriptions of the sites are provided below, as the detailed descriptions are in the supplement (S1).

The ECCC continuous measurement of atmospheric $CH_4$ started in the late 1980s, at Alert Observatory (ALT, 82.5˚ N, 62.5˚ W) in 1987, followed by the measurement at Fraserdale (FSD, 49.9˚ N, 81.6˚ W) in 1989. Alert was established to monitor the
baseline GHG for the pan-Arctic region, while Fraserdale has been monitoring $CO_2$ and $CH_4$ in the northern wetlands and boreal forest ecosystems. In the 2000s, the ECCC measurement program was gradually expanded from the west to the east across Canada: Estevan Point (ESP, 49.4˚ N, 126.5˚ W) on the Pacific coast, and continental sites, Lac La Biche (LLB, 54.9˚ N, 112.5˚ W), and East Trout Lake (ETL, 54.4˚ N, 104.9˚ W), Egbert (EGB, 44.2˚ N, 79.8˚ W), Chibougamau (CHM, 49.7˚ N, 74.3˚ W) which was later replaced by Chapais (CPS, 49.8˚ N, 74.9˚ W), and Sable Island (WSA, 43.9˚ N, 60.0˚ W) off the
Atlantic coast. In the 2010s, the observation network was further expanded to subarctic region in northern Canada, Inuvik



(INU, 68.3˚ N, 133.5˚ W), Behchoko (BCK, 62.8˚ N, 115.9˚ W), and Churchill (CHL, 58.7˚ N, 93.8˚ W), and also to populated areas of southern Canada, Downsview (DWS, 43.8˚ N, 79.5˚ W) in Toronto.

**Table 1.** Measurement sites. North, West, East and South indicate the four subregions defined in Sect. 2.2.4 (see Fig. 4a).

|  | Site | ID | Latitude | Longitude | Elevation (m) | Intake height (m) |
|---|---|---|---|---|---|---|
| North | Alert | ALT | 82.5˚ N | 62.5˚ W | 200 | 10 |
|  | Inuvik | INU | 68.3˚ N | 133.5˚ W | 113 | 10 |
|  | Behchoko | BCK | 62.8˚ N | 115.9˚ W | 160 | 60 |
| West | Estevan Point | ESP | 49.4˚ N | 126.5˚ W | 7 | 40 |
|  | Lac La Biche | LLB | 54.9˚ N | 112.5˚ W | 540 | 50 |
|  | East Trout Lake | ETL | 54.4˚ N | 104.9˚ W | 493 | 105 |
| East | Churchill | CHL | 58.7˚ N | 93.8˚ W | 29 | 60 |
|  | Fraserdale | FSD | 49.9˚ N | 81.6˚ W | 210 | 40 |
|  | Chibougamau | CHM | 49.7˚ N | 74.3˚ W | 393 | 30 |
|  | Chapais | CPS | 49.8˚ N | 74.9˚ W | 391 | 40 |
|  | Sable Island | WSA | 43.9˚ N | 60.0˚ W | 5 | 25 |
| South | Egbert | EGB | 44.2˚ N | 79.8˚ W | 251 | 25 |
|  | Downsview | DWS | 43.8˚ N | 79.5˚ W | 198 | 20 |


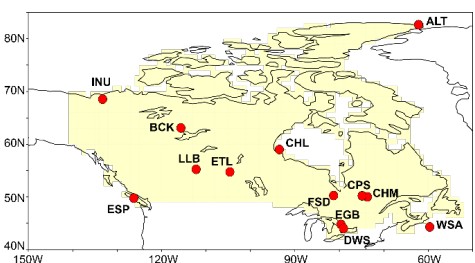

**Figure 1. The ECCC atmospheric measurement sites used in this study**.

Atmospheric $CH_4$ measurements were initially made using a gas chromatograph (GC, Hewlett Packard HP5890 or HP6890) with flame ionization detection (FID). From 2013 onwards, GC systems were gradually replaced with cavity ring-down spectrometers (CRDS, Picarro G1301, G2301, or G2401). All measurements are traceable to the World Meteorological





Organization X2004 scale (Dlugokencky et al., 2005). A detailed description of the $CH_4$ measurement system can be found elsewhere (e.g., Chan et al., 2020).


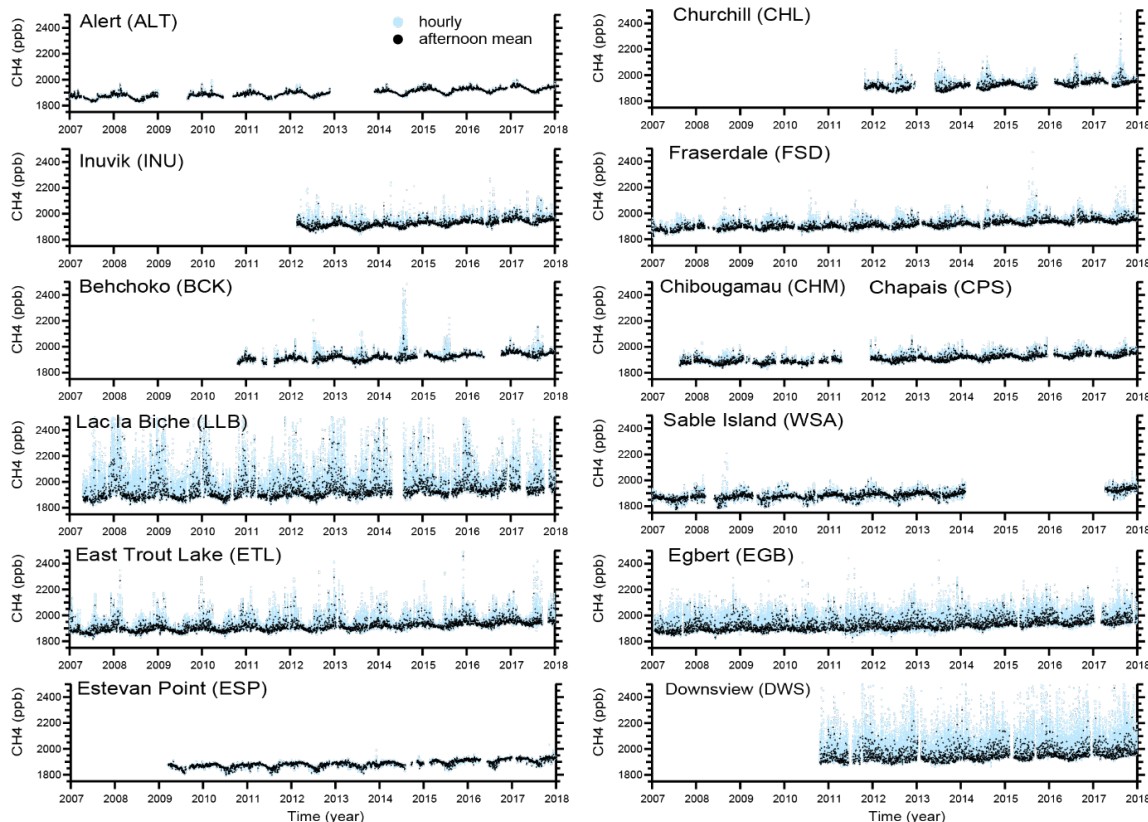

**Figure 2. Time series of atmospheric $CH_4$ mixing ratios at Canadian sites. The observed values are shown as the hourly means (light-blue dot) and afternoon mean (black dot, 12:00–16:00 local time) from continuous measurements.**


After the initial quality control, all the atmospheric $CH_4$ measurements are reduced to hourly mean values. To minimize the impact of local sources on the regional-scale $CH_4$ flux estimates, the hourly data are averaged to afternoon means (from 12:00 to 16:00 local time), assuming midday air is in a well-mixed planetary boundary condition. Then, a curve-fitting method is applied to the time series to remove the outliers from the measurements, which indicate contamination from unknown sources.

The curve-fitting method has two harmonics of 1-year and 6-month cycles and two low- and high-pass digital filters with cut-off periods of four months and 24 months (Nakazawa et al., 1997). The threshold of outliers is set to be three times the standard deviation of the residual of the best-fit curves. Figure 2 shows the hourly and afternoon mean atmospheric $CH_4$ time



series at the 13 measurement sites. The respective time series show different variations, reflecting their local and subregional CH$_4$ source strengths.

## 2.2 Regional inverse model

In this study, we used the Bayesian inversion approach to estimate the regional CH$_4$ fluxes over Canada. The same inverse modelling framework was previously used for CH$_4$ flux estimation in the Canadian Arctic (Ishizawa et al., 2019). The Bayesian inversion optimizes fluxes to minimize the differences between the observations and the modelled atmospheric CH$_4$ mixing ratios. This study calculated modelled CH$_4$ mixing ratios based on the backward runs of Lagrangian particle dispersion models (LPDMs). The optimized flux uncertainties from modelling errors were estimated from an ensemble of 24 inversion experiments using multiple transport models and prior flux estimates. The following sections describe the regional inverse model.

### 2.2.1 Regional inversion

The Bayesian inversion optimizes the scaling factors of posterior fluxes by minimizing the mismatch between modelled and observed mixing ratios with constraints and given uncertainties using the cost function ($J$) minimization method (Lin et al., 2004):

$$J(\boldsymbol{\lambda}) = (\boldsymbol{y} - \boldsymbol{K}\boldsymbol{\lambda})^T \boldsymbol{D}_\epsilon^{-1} (\boldsymbol{y} - \boldsymbol{K}\boldsymbol{\lambda}) + (\boldsymbol{\lambda} - \boldsymbol{\lambda}_{prior})^T \boldsymbol{D}_{prior}^{-1} (\boldsymbol{\lambda} - \boldsymbol{\lambda}_{prior}), \tag{1}$$

where $\boldsymbol{y}$ (N×1) is the vector of observations (To be comparable to the modelled CH$_4$ (denoted as $\boldsymbol{K}\boldsymbol{\lambda}$) based on the prior fluxes, the background mixing ratio representing the CH$_4$ signal from five days prior to the observation time has been subtracted from the observed mixing ratios, see the following Sect. 2.2.2). N is the number of time points times number of stations (N is reduced if observations are missing). $\boldsymbol{\lambda}$ (R×1) is the vector of the posterior scaling factors to be estimated, and R is the number of subregions to be solved. $\boldsymbol{\lambda}_{prior}$ is the vector of the prior scaling factors which are all initialised to 1 for subregions, and $\boldsymbol{K}$ (N×R) is the matrix of contributions from R subregions. $\boldsymbol{K}$ is a Jacobian matrix of flux sensitivity, a product of two matrices, $\boldsymbol{M}$ and $\boldsymbol{x}$. $\boldsymbol{M}$ is the modelled transport (or footprints in this study), and $\boldsymbol{x}$ is the spatial distribution of the surface fluxes. A linear regularisation term has been added, which is the second term on the right-hand side of the equation. $\boldsymbol{D}_\epsilon$ and $\boldsymbol{D}_{prior}$ are the error covariance matrices. $\boldsymbol{D}_\epsilon$ is the prior model-observation error/uncertainty matrix (N×N) where the diagonal elements are $(\sigma_e)^2$. $\boldsymbol{D}_{prior}$ is the prior scaling factor uncertainty matrix (R×R) where the diagonal elements are $(\sigma_{prior})^2$. The model-observation mismatch errors are treated as uncorrelated to each other and the contributions from the subregions are also uncorrelated. All the off-diagonal elements in $\boldsymbol{D}_\epsilon$ and $\boldsymbol{D}_{prior}$ are assumed to be zero. We assigned $\sigma_e$= 0.33 for the model-observation error and $\sigma_{prior}$= 0.30 for the prior uncertainty (Ishizawa et al., 2019). The inversion's sensitivity to these uncertainties was examined by doubling their values. The results show the optimized fluxes are not strongly dependent on these prescribed uncertainties. The estimate for $\boldsymbol{\lambda}$ is calculated according to the expression below (Lin et al., 2004):



$$\lambda = \left(K^T D_\epsilon^{-1} K + D_{prior}^{-1}\right)^{-1}\left(K^T D_\epsilon^{-1} y + D_{prior}^{-1}\lambda_{prior}\right). \tag{2}$$

The posterior error variance-covariance, $\Sigma_{post}$, for the estimates of $\lambda$ is calculated:

$$\Sigma_{post} = \left(K^T D_\epsilon^{-1} K + D_{prior}^{-1}\right)^{-1}. \tag{3}$$

We optimize the total $CH_4$ fluxes, including all the $CH_4$ fluxes on a monthly time resolution.

### 2.2.2 Atmospheric models

In an LPDM, air-following particles travel backward from the measurement location at a given initiation time (corresponding to the time of observation) and provide the relationship between surface fluxes and atmospheric mixing. This relationship is
called footprint, source-receptor relationship, or flux sensitivity. To estimate the transport model errors in the flux estimate, three different models were employed in this study, combining two different LPDMs, FLEXible PARTicle dispersion model (FLEXPART) (Stohl et al., 2005) and Stochastic Time-Inverted Lagrangian Transport Model (STILT) (Lin et al., 2003; Lin and Gerbig, 2005), and three different meteorological data sets. These three model setups are here named FLEXPART_EI, FLEXPART_JRA55 and WRF-STILT. FLEXPART_EI is FLEXPART v8.2 driven by the European Centre for Medium-
range Weather Forecast (ECMWF) ERA-Interim (Dee et al., 2011; Uppala et al., 2005), FLEXPART_JRA55 is FLEXPART v8.0 driven by the Japanese 55-year Reanalysis (JRA-55) from Japanese Meteorological Agency (JMA) (Kobayashi et al., 2015), and WRF-STILT is STILT driven by Weather Research and Forecasting (WRF) model (e.g., Hu et al., 2019). The WRF-STILT footprints used in this study were provided by the NOAA CarbonTracker-Lagrange project (CT-L, https://gml.noaa.gov/ccgg/carbontracker-lagrange). All the footprints calculated by the respective models were mapped into
$1.0°\times1.0°$.

LPDMs simulate surface contributions for a certain period prior to the measurements at sites by air-following particles. In this study, at the endpoints of the particles after 5-day back-trajectory, the background conditions of atmospheric $CH_4$ mixing ratios were provided by a global model, National Institute for Environmental Studies Transport Model (NIES TM) with optimized global $CH_4$ flux fields by the GELCA $CH_4$ inverse model (Ishizawa et al., 2016). The performance of NIES TM simulation
with GELCA-$CH_4$ optimized fluxes was reported in Chan et al. (2020).

### 2.2.3 Prior CH₄ fluxes

We considered eight scenarios of prior emissions, combining four different wetland fluxes and two anthropogenic emission inventories (Table 2).

The first wetland ensemble model, WetCHARTs, derives wetland $CH_4$ fluxes as a function of a global scaling factor, wetland
extent, heterotrophic respiration and temperature dependence (Bloom et al., 2017). We used the ensemble mean fluxes over 18 model sets, available for 2001-2015. The second wetland flux set is the monthly climatological estimates from the ensemble



mean of 16 wetland process-based models (Poulter et al., 2017), which was provided for the GCP-CH$_4$ inversion project (Saunois et al., 2020) (GCPwet in short hereinafter). The last two wetland CH$_4$ fluxes are from the Canadian land Surface Scheme including Biogeochemical Cycles (CLASSIC), which is a successor to the Canadian Land Surface Scheme (CLASS)

and the Canadian Terrestrial Ecosystem Model (CTEM) (Melton et al., 2020). The four sets of CLASSIC wetland CH$_4$ fluxes were calculated with two different meteorological datasets, CRU-JRA (Harris et al., 2020) and GSWP3 (Dirmeyer et al., 2006), which use diagnostically specified and prognostically determined wetland extents. These two different schemes predict different spatial distribution and temporal variations of wetland CH$_4$ emissions. The choice of the meteorological data set appears to be less influential to simulated CLASSIC CH$_4$ fluxes, indicating that the model response to both meteorological

forcings is consistent. Therefore, these four sets were aggregated to the two sets of CLASSIC wetland CH$_4$ fluxes (diagnostic and prognostic wetland extents), by averaging simulated CH$_4$ emissions for the two different meteorological forcing data sets. These diagnostic and prognostic CLASSIC CH$_4$ flux sets are abbreviated as CLASSICdiag and CLASSICprog, respectively. Figure 3a shows the spatial distribution of the four wetland CH$_4$ fluxes for a summer month (July) and a winter month (January). Overall GCPwet shows stronger emissions than the other estimates, especially along Hudson Bay and around the

border between Northern Territories and Alberta in western Canada, resulting in larger annual emissions at subregional and national levels (Fig. 3c). GCPwet shows strongest summer emission, but weaker winter emissions than WetCHARTs and CLASSICdiag, which have tangible winter emissions. CLASSICprog also shows almost negligible winter emissions, while summertime emissions are stronger than CLASSICdiag (Fig. 3e). Annually the two CLASSIC wetland fluxes, CLASSICdiag and CLASSICprog, have similar annual emissions at the national and subregional levels (Fig. 3c).

**Table 2.** Prior fluxes used in this study. Eight prior flux scenarios were made as combinations of two anthropogenic fluxes (EDGAR, ECAQ) and four wetland fluxes (WetCHARTs, GCPwet, CLASSICdiag, and CLASSICprog). For other natural fluxes, the same prior datasets were used in all the scenarios.

| Category | | Abbrev. in this study | Dataset |
|---|---|---|---|
| Anthropogenic | Energy, Agriculture, Waste | EDGAR | EDGAR v4.3.2 |
| | | ECAQ | ECCC-AQ2013 |
| Natural | Wetland | WetCHARTs | WetCHARTs v1.0 (Bloom et al., 2017) |
| | | GCPwet | Poulter et al. (2017); Saunois et al. (2020) |
| | | CLASSICdiag | CLASSIC diagnostic wetland CH$_4$ |
| | | CLASSICprog | CLASSIC prognostic wetland CH$_4$ |
| | Soil uptake | | VISIT (Ito and Inatomi, 2012) |
| | Biomass burning | | GFASv1.2 (Kaiser et al., 2012) |





**Figure 3.** Wetland and anthropogenic prior CH4 fluxes in this study. Spatial distributions (a, b), annual fluxes (c, d) and seasonal cycles (e, f) for Canada and subregions. The variations of wetland CH4 fluxes are climatological or multi-year means, and those of anthropogenic CH4 fluxes are for 2013. Subregions, North, West, East and South, are defined in Sect. 2.2.4 (see Fig. 4a).



The two sets of anthropogenic CH₄ emissions used in this study are the monthly ECCC-AQ2013 (ECAQ) scaled to the NIR sectoral totals for year 2013 by province (Chan et al., 2020), and the annual Emissions Database for Global Atmospheric Research (EDGAR) v4.3.2 (Janssens-Maenhout et al., 2019). As seen in Fig. 3b, Canada's anthropogenic emissions are concentrated around the western provinces and the southern border. The spatial patterns of both prior emission datasets are quite similar, though there are some differences in hotspot locations (Figs. 3b and 3d). The seasonal variability in ECAQ is
small compared to the variability in wetland fluxes (Figs. 3e and 3f).

The CH₄ emission estimates for two other categories used in this study are, daily biomass burning (BB) emissions from Global Fire Assimilation System (GFAS) v1.2 (Kaiser et al., 2012), and the climatological monthly soil CH₄ uptake based on VISIT-CH₄ (Ito and Inatomi, 2012). In Canada, the VISIT-modelled soil uptake is weak (-1.6 Tg CH₄ year⁻¹) and widely distributed spatially in the summer months. If the prior data do not cover the whole analysis period, the respective last 5-year mean values
were repeatedly used afterwards, climatological prior emission datasets (e.g., GCPwet) were used without further processing. All data are converted into 1.0°×1.0° by simply aggregating finer spatial resolution data (e.g., EDGAR) or re-gridding coarser data (e.g., CLASSIC).

**2.2.4 Domain and subregions**

The regional domain of interest for this study is Canada. We set up two subregion masks for Canada, mainly based upon
climate zone with provincial/territorial division and industrial activities also considered (see Fig. S2 for the Canadian provinces/territories). Outside Canada was treated as one outer region. The first mask consists of four subregions, North, West, East and South (Fig. 4a). The second mask reduces the four subregions to two, North and West become West_2 and East and South become East_2 (Fig. 4b). The subregion North covers the Canadian Arctic, including Northwest Territories (NT), Yukon (YT), and Nunavut (NU). In North, the major CH₄ sources are natural, primarily wetlands and occasionally some biomass
burning. The subregion West includes the three western provinces, British Columbia (BC), Alberta (AB) and Saskatchewan (SK). AB and SK are the largest oil and gas CH₄ emitting provinces in Canada, which account for ~70% of the national emissions from the oil/gas sectors (ECCC, 2022).   The subregion West has other anthropogenic sources such as the agriculture sector and natural sources, primarily wetlands. The CH₄ emission in the subregion East is mainly from natural wetlands, where the Hudson Bay Lowlands (HBL) are located. The subregion South is southern Ontario (ON) and Quebec (QC) (south of 48°
N), the most populated area in Canada, resulting in considerable anthropogenic CH₄ emissions from energy and agriculture sectors and landfills (waste management).

The sensitivity of the flux estimation to the number of subregions was examined; increasing the spatial resolution to six subregions revealed some model instability problems. With the limited observational constraints during this study period, subregions with weak prior fluxes or lack of measurement sites demonstrated larger uncertainties in estimated fluxes with



frequent unrealistic negative fluxes. Similar relations on resolving power of inversions were reported in Ishizawa et al. (2019)
       and Chan et al. (2020). Therefore, this study focuses on the inversion results of the four subregions and the two larger
       subregions.

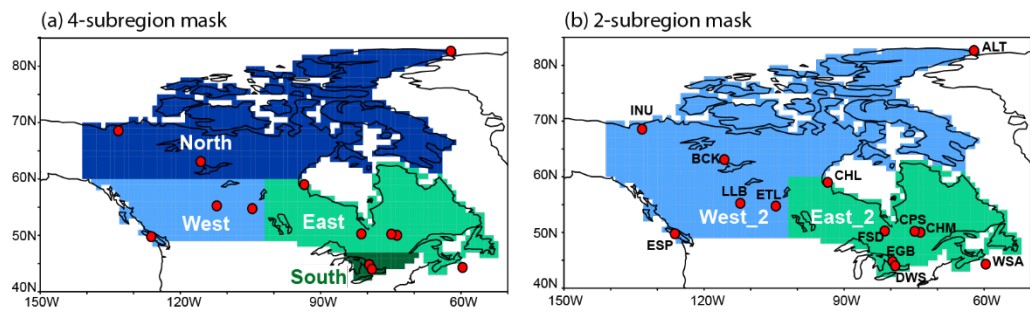


**Figure 4. Subregion masks for inversion and the ECCC atmospheric measurement sites. Canada is divided into (a) four subregions
(North, West, East and South) and (b) two subregions (West_2, East_2). These subregion masks are based on Canadian provinces
and territories (see Fig. S2), climate zones and industrial activities.**

**2.2.5 Experimental setup**

       Figure 5 shows the schematic diagram of the inversion experiments regarding the combinations of prior fluxes, transport
       models, subregion masks and observations. The ensemble of 24 experiments consists of the permutations of eight prior flux
       scenarios and three transport models, as summarized in Table S1.   We performed these 24 experiments with the 4-subregion
       mask and all 12 site observations as the reference inversion (abbreviated as Inv_4R12S). As a sensitivity test to examine the
impact of on observational coverage, two additional inversions using the 2-subregion mask with the 12 sites (Inv_2R12S) and
       two sites of ETL and FSD (Inv_2R2S) were conducted with the same ensemble setup of 24 experiments. ETL and FSD have
       long measurement records extending back beyond the period of this study. Therefore, the inversion Inv_2R2S explored the
       feasibility of estimating $CH_4$ fluxes by inversion for a longer time period.





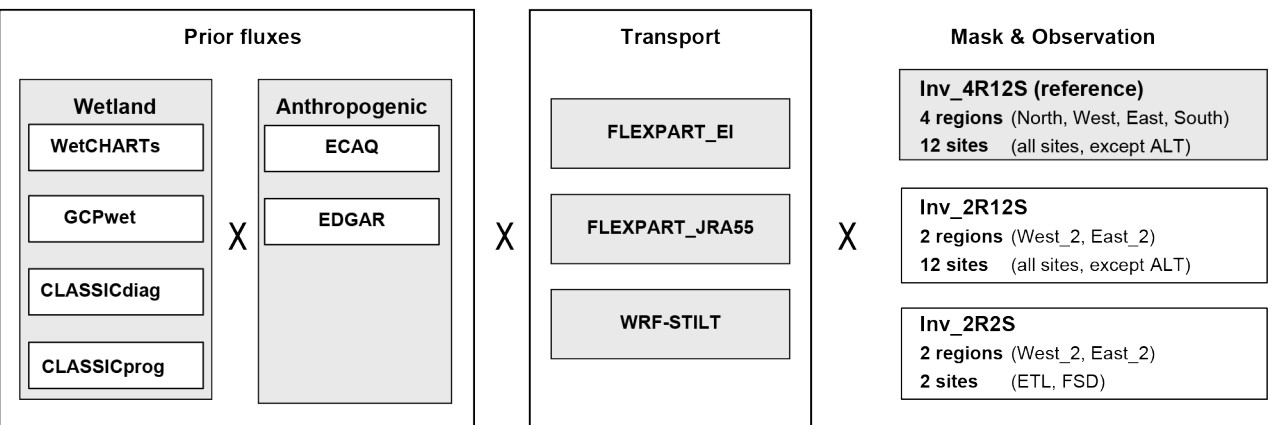

**Figure 5. Diagram of inversion experiment settings in this study. Eight prior emission scenarios out of four wetland fluxes and two anthropogenic fluxes are applied to three different transport models. These flux-transport combinations yield the 24 experiments as listed in Table S1. The experiments with a 4-subregion mask and 12 observation sites are conducted as the reference inversion (Inv_4R12S). The experiments with a 2-subregion mask and 12 observation sites (Inv_2R12S) or two sites (Inv_2R2S) are performed as additional inversions. The mask maps are defined in Fig. 4, along with observation site locations.**

## 2.3 Partition into natural fluxes and anthropogenic fluxes

The posterior $CH_4$ flux in this study is the total flux (natural plus anthropogenic). Thus, we need a scheme with assumptions to partition the total fluxes into natural and anthropogenic sources. Some previous studies for the northern extra-tropical region (e.g., Tohjima et al., 2014; Thompson et al., 2017), estimated the anthropogenic sources by assuming that, in the winter season, anthropogenic $CH_4$ fluxes are dominant while biogenic $CH_4$ fluxes (e.g., natural wetlands or rice cultivation) fluxes are dormant and negligible. This assumption is consistent with many process-based wetland models, as demonstrated by the prior wetland fluxes used in this study (see Fig. 3e). In this study, the prior wetland $CH_4$ winter flux fraction (November to March, < 60˚N) to annual emission is in the range of 2.6 % to 9.2 %.

Here, taking into account the possible winter wetland $CH_4$ emissions into the flux partition, we applied the following simple scheme to partition natural fluxes into warm (growing) and cold (non-growing) seasons, with the assumption of temporally uniform anthropogenic emissions. Let $f_{total}(m)$ be the monthly posterior total flux, and $F_{total}$ be the annual total flux. Then, $F_{total}$ consists of annual natural and anthropogenic fluxes ($F_{natural}$, $F_{anthropogenic}$), and the annual total flux could also be expressed as the sums of monthly total fluxes in the warm season ($F_{total\_warm}$) and the cold season ($F_{total\_cold}$):





$$F_{total} = \sum_{m=1}^{12} f_{total}(m)$$

$$= F_{natural} + F_{anthropogenic}, \tag{4}$$

$$F_{total} = \sum_{m=warm} f_{total}(m) + \sum_{m=cold} f_{total}(m)$$

$$= F_{total\_warm} + F_{total\_cold.} \tag{5}$$

Next, these annual fluxes could be expanded in terms of the fraction, $R_{cold}$, of cold season's natural $CH_4$ emissions to its annual emission and the number of cold months, $N_{cold}$, as in Eqs. (6) and (7):

$$F_{total\_cold} = F_{natural\_cold} + F_{anthropogenic\_cold}$$

$$= R_{cold} \times F_{natural} + \left(\frac{N_{cold}}{12}\right) \times F_{anthropogenic} \tag{6}$$

$$F_{total\_warm} = F_{natural\_warm} + F_{anthropogenic\_warm}$$


$$= (1 - R_{cold}) \times F_{natural} + \left(1 - \frac{N_{cold}}{12}\right) \times F_{anthropogenic}, \tag{7}$$

where

$$R_{cold} = \frac{F_{natural\_cold}}{F_{natural}},$$

and $N_{cold}$ is number of months in cold season.

If $R_{cold}$ and $N_{cold}$ for each subregion are given, the annual fluxes of natural and anthropogenic, $F_{natural}$, $F_{anthropogenic}$ can be solved through Eqs. (6) to (7). We define the cold (non-growing) season as November to March for all the subregions (< 60˚N), except from October to May for North, Canadian Arctic (> 60˚ N) following Treat et al. (2018). The cold season approximates the period when the air temperature is below 0˚C, as seen in Fig. 10. The estimation of $R_{cold}$ is explained in Sect. 3.5.2.

## 3 Results and Discussion

### 3.1 Estimated monthly CH4 fluxes

The monthly posterior $CH_4$ fluxes from 2007 to 2017 for the four subregions and national total are shown in Fig. 6 and Fig. S3. The posterior fluxes during the early period from 2007 to 2011 are highly variable, most notably in North, with posterior fluxes showing unrealistic negative fluxes indicating they are not constrained by the inverse model. Before 2012, the subregion North did not have sufficient observations to constrain the inverse model (see Fig. 4a). The marine boundary layer site, Alert

(ALT) at the northern end of the subregion North, appears not to see the subregional flux signals (mainly in the southern part of the subregion) above the background atmospheric $CH_4$ (Ishizawa et al., 2019). Therefore, ALT was not used in this study,



following the inversion study of Canadian Arctic CH₄ (Ishizawa et al., 2019). Similarly, the measurement sites far to the south (e.g., Lac La Biche (LLB), East Trout Lake (ETL) in West) also could not constrain the fluxes in North; the stronger flux signals in West tend to mask the signals from North. In October 2010, the site Behchoko (BCK) in the southern part of North

began measurements. However, the presence of data gaps still likely resulted in negative fluxes in North for 2011. From 2012 onward, BCK, along with the new sites Inuvik (INU) and Churchill (CHL), provided sufficient constraints on North to yield positive (more reasonable) posterior fluxes with a summer maximum in the seasonal cycle from the wetland CH₄ fluxes.

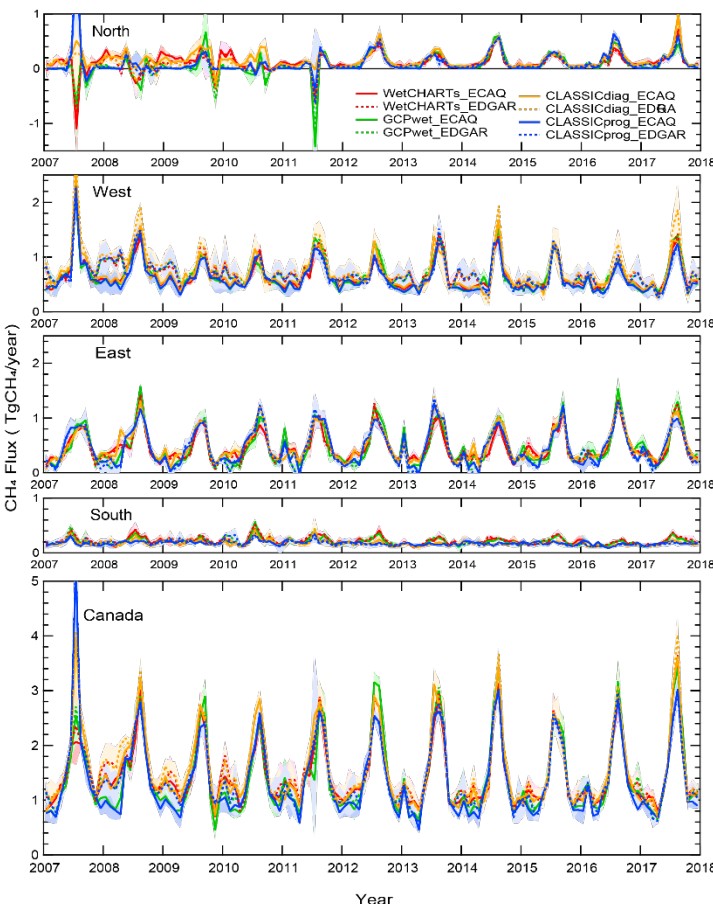

**Figure 6. Monthly posterior CH₄ fluxes for sub-regions and Canada in reference inversion, Inv_4R12S. The lines are the mean**
**posterior fluxes of experiments with three transport models (FLEXPART_EI, FLEXPART_JRA55, and WRF-STILT) per prior flux scenario. The eight prior flux scenarios are used: WetCHARTs_ECAQ (red solid), WetCHARTs_EDGAR (red dotted), GCPwet_ECAQ (green solid), GCPwet_EDGAR (green dotted), CLASSICdiag_ECAQ (orange solid), CLASSICdiag_EDGAR (orange doted), CLASSICprog_ECAQ (blue solid) and CLASSICprog_EDGAR (blue dotted). The respective shaded areas indicate the range of minimum and maximum estimates.**




The subregion West (on the south side of the subregion North, see Fig. 4a) also shows more variability in the posterior fluxes before 2012, particularly in the 2008 and 2010 winters. The presence of the poorly constrained North before 2012 (an extra degree of freedom in the inversion) appears to influence the statistical optimization of the inverse model as a whole, leading to more variability in the posterior fluxes in West. The statistical nature of the inverse model does not always yield physically

realistic solutions. Thus, it is important to evaluate the setup and results of the inverse model for physical consistency before making other interpretations of the inversion results. As noted from 2012 onward, there appear to be sufficient sites to constrain North. Consequently, the posterior fluxes for West also show less variability or more robustness after 2012.

Figure 6 also shows the variability in the seasonal cycle of posterior fluxes among the inversions and the inter-annual variability in the magnitudes, particularly in the summer maxima in the different subregions. Section 3.4.1 discusses the variability in the

seasonal cycle of the fluxes among the different inversion settings, as well as the ensemble mean seasonal cycles for subregions. The inter-annual variation of summer wetland fluxes and the relationship to meteorological conditions are examined in Sect. 3.4.2.

**3.2 Trend of annual mean fluxes**

In Sect. 3.1, the inversion results with four subregions and 12 observation sites (reference inversion, Inv_4R12S) show large

variability and even non-physical negative $CH_4$ fluxes for some subregions in the early period (2007−2011). The potentially poorly constrained fluxes could affect the trends in the results. Thus, we performed the two additional inversions with different settings. The first inversion (Inv_2R12S) used two subregions (West_2 and East_2 in Fig. 4b) and constrains the fluxes by the same set of observations with the reference inversion (Inv_4R12S); the second inversion (Inv_2R2S) used the same two subregions, and constrained the fluxes by two measurement sites, ETL and FSD. The advantage of using fewer subregions is

having more observations per subregion to constrain the fluxes; the disadvantage is the inability to estimate the possible differences within each subregion. These inversions measure the stability or robustness (to changing setups) of the inversion results, including the possible trend. The sites ETL and FSD are the sites that have the longest records in the mainland of Canada, covering the entire period of this study. As ETL and FSD are located in western and eastern Canada respectively, these two sites could potentially constrain the subregions, West_2 and East_2, respectively. The respective and combined

footprints for ETL and FSD are shown in Fig. S4.

To investigate the presence of trends over 2007−2017, the mean annual posterior $CH_4$ fluxes (ensemble means of 24 experiments per inversion setup as described in Fig. 5) are shown for all of Canada and two large subregions (West_2 and East_2) from the three inversion setups (Inv_4R12S, Inv_2R12S and Inv_2R2S) together in Fig. 7. Fig. S5 shows the mean annual posterior $CH_4$ fluxes by prior flux scenario per inversion setup. For the whole inversion period from 2007 to 2017,





these three inversions agree within the range of results among the 24 experiments per inversion setup (shown as the shaded bands) for Canada and the two large subregions. For the later period with more observational coverage (2012−2017), the three inversions are in better agreement. Thus, the estimated fluxes appear robust to the different setups used for the whole period. However, the variability of the inversion results or shaded bands appears larger in the beginning period when the observational coverage was limited. In addition, the inversion Inv_2R2S has larger inter-annual variability than the inversions constrained by 12 sites. This is consistent with the statistical nature of the Bayesian inversion; statistical inferences are generally better with larger samples of data or observations.

Comparing all the inversion results for long-term trends for the 11 years (Fig. 7), there is no consistent trend for Canada and the two subregions. The mean trend slopes and uncertainties are shown in the supplement, Table S2. The inversion Inv_2R2S shows a slight downward trend in East_2, but the trend is within the inter-annual variability in the estimated fluxes. This possible trend is not replicated in the other two inversion setups using 12 observation sites (Inv_4R12S, and Inv_2R12S). The apparent trend may be due to insufficient observations (making the results sensitive to missing observations) to statistically constrain the fluxes. As shown in Fig. S1, the data availability at FSD is low (< 10 per month) for four months, March to June 2008, possibly resulting in less constrained fluxes for East_2 in Inv_2R2S. In the inversions using 12 observation sites (Inv_4R12S and Inv_2R12S), CHM, which is ~500 km east of FSD, provides full observational constraints for those four months. Another caveat in trend analysis for subregions is exhibited in Fig. S6 for Inv_4R12S. With the four subregions, the subregions North and West (equivalent to West_2) are showing upward and downward trends, respectively, which is possibly another signal of the lack of observational constraints in the early period noted above. There are opposite trends in North and West, resulting in the absence of a trend (not a statistically significant trend) for the combined subregion West_2, as shown in Fig. 7 and in the supplement, Fig. S6 and Table S2. This result indicates that sparse data coverage could yield spurious trends on the subregional scale from top-down analysis.

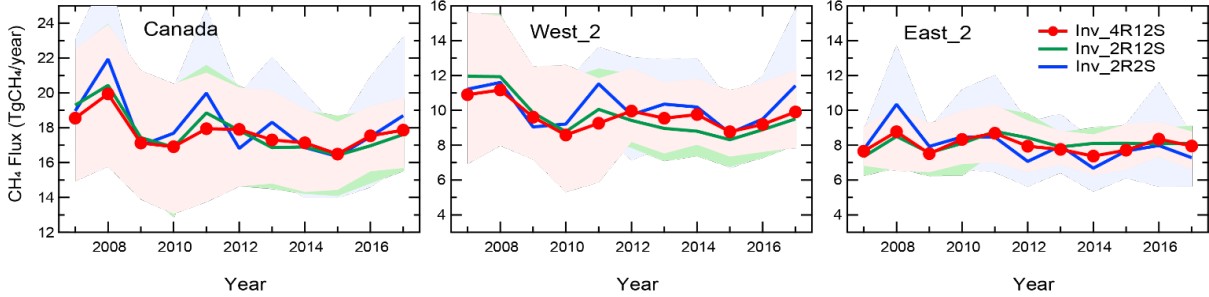

**Figure 7. Trend of estimated yearly CH₄ fluxes in Canada and western (West_2) and eastern (East_2) subregions from three inversion setups, 72 experiments in total. Lines show mean fluxes over each of three inversion sets with different subregion masks and observation site selections. The shaded areas indicate the range of maximum and minimum estimates among 24 experiments per inversion setup.**



Our result of no significant long-term trend for the national total $CH_4$ emissions during the period 2007—2017 contrasts with some other studies showing possible trends. For example, Thompson et al. (2017), from the 9-year inversion for 2005—2013, concluded that a positive trend in $CH_4$ emissions in North America (> 50˚N, Canada and Alaska) of 0.38 to 0.57 Tg $CH_4$ year[-2], especially in the HBL due to warming soil temperature. The ensemble mean of GCP-$CH_4$ global inversions showed a gradual downward trend over the last two decades of 2000–2017 (Stavert et al., 2021), which is attributed to a reduction of wetland emissions at ~-0.3 Tg $CH_4$ year[-2]. However, the inferred downward trend does not agree with the ensemble of the process-based wetland model estimate; all the wetland models show an upward trend (Stavert et al., 2021). These contrasting results point to the difficulty of inferring long-term trends from highly variable data and insufficient data coverage for Canada. Thus, continuous atmospheric $CH_4$ measurements with good spatial coverage are needed to detect any long-term changes in $CH_4$ emissions in response to climate forcings or anthropogenic emission changes.

### 3.3 Evaluation of posterior fluxes

Model-data comparison of atmospheric mixing ratio at the measurement sites is commonly employed to evaluate the posterior fluxes. Figure S7 shows the model-data comparison by measuring the mean biases and correlation coefficients between the simulated and observed mixing ratios at each site in all 24 experiments for the reference inversion Inv_4R12S. The results of the simulated prior mixing ratios are overall dependent on the prior fluxes and transport models. The simulated posterior mixing ratios show an improvement in matching with the observations at most of the sites, except ESP. Also, DWS exhibits notable transport model dependency. The FLEXPART_JRA cases show larger biases than the other transport model cases. This might be related to the resolution of the driving meteorological data of FLEXPART_JRA (1.25˚×1.25˚ on horizontal resolution and 6-hourly time step) than other models (1.0˚×1.0˚ and 3-hourly in FLEXPART_EI and 10 km×10 km and 1-hourly in WRF-STILT), which might be too coarse to model the urban site DWS. The correlations between the posterior mixing ratios and the observations are improved, being around 0.9 at all the sites. The correlation at ESP is unchanged after the inversion. This indicates the ESP, which is the most western site on the Pacific coast in Canada, has already been assimilated well by the background mixing ratios, and not strongly influenced by the continental fluxes. On the other side of the continent, the most eastern site WSA, on Sable Island in the Atlantic, shows a slight improvement after the inversion.

In this study, we explored another type of data for posterior flux evaluation. There is flux information in the observed mixing ratio difference (or gradient) between sites. Fan et al. (1998) noted that the downwind and upwind mixing ratios' difference for a given region should reflect the source/sink strength within the region. For example, for $CO_2$ uptake of 1.7 Gt C year[-1] over North America, there could be an annual difference of ~0.3 ppm from the Atlantic coast to the Pacific coast. Such small differences are challenging to extract, given the large variability in the atmospheric $CO_2$ mixing ratios. However, on smaller spatiotemporal scales (mesoscale to microscale), local or urban emission studies (e.g., Bréon et al., 2015; Mitchell et al., 2018),





and large-point source estimates from satellites (e.g.,Nassar et al., 2022) have used mixing ratio differences to constrain city scale or facility scale emissions. Thus, the following examines the relationship between mixing ratio difference and regional fluxes on larger (synoptic) scales of this study.

405  For the case of the mixing ratio difference ($\Delta C_{ETL-FSD} = C_{ETL} - C_{FSD}$) between East Trout Lake (ETL) and Fraserdale (FSD), which are ~2600 km apart and approximately along the prevailing westerly wind direction, there are no consistent correlations between $\Delta C_{ETL-FSD}$ and $C_{ETL}$, or between $\Delta C_{ETL-FSD}$ and $C_{FSD}$ (see Fig. S8 in the supplement). As the uncorrelated information in $\Delta C_{ETL-FSD}$ has not been used to constrain the inverse model, $\Delta C_{ETL-FSD}$ could serve as an evaluation of the inversion results. The comparison of multi-year (2012–2017) averaged monthly mixing ratio difference between models and observations is

410  shown in Fig. 8. In addition to comparing the east-west difference ($\Delta C_{ETL-FRD}$ between ETL and FSD), the north-south difference ($\Delta C_{EGB-FRD} = C_{ETL} - C_{FSD}$) between Egbert (EGB) and FSD is also compared in Fig. 8. The posterior annual mean correlation coefficients are improved, being computed as 0.6 for $\Delta C_{ETL-FRD}$ (0.4 for the prior) and 0.7 for $\Delta C_{EGB-FRD}$ (0.5 for the prior).

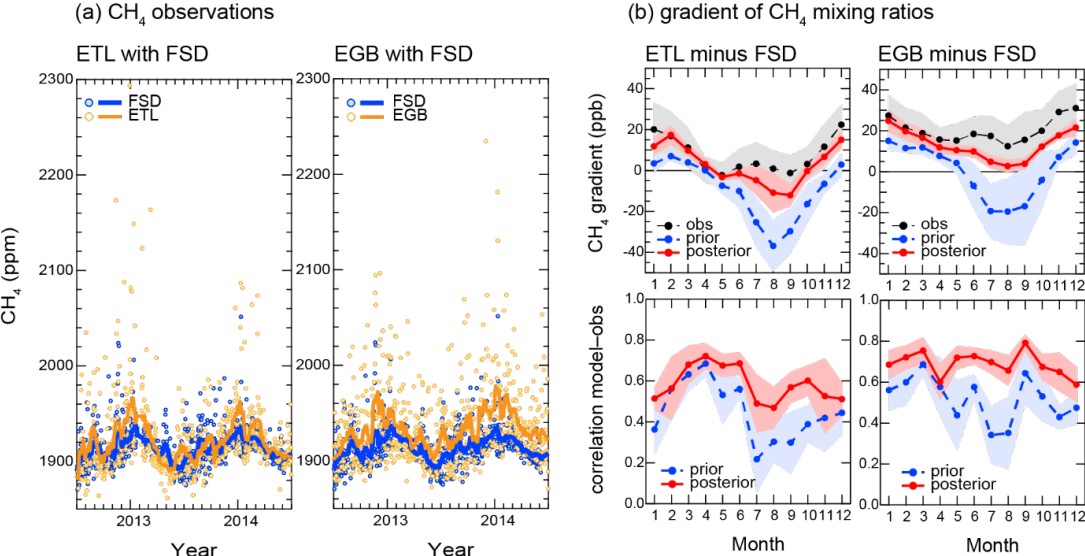

**Figure 8. (a) Examples of time series of CH₄ mixing ratios observed at ETL and EGB, along with the CH₄ mixing ratios at FSD. The circles are afternoon means, and solid lines are smoothed curves with a 20-day moving window average. (b) Gradients of mixing ratio between ETL and FSD, EGB and FSD (top) and the correlation (bottom). In the top, red solid lines with markers are the mean gradients of modelled CH₄ mixing ratios with posterior fluxes over 2012–2017, and blue dotted lines with markers are the modelled gradients with prior fluxes. Black lines with markers are mean gradients of observed CH₄ mixing ratios. At the bottom, red lines**

420  **show the mean correlation between the modelled posterior gradient and observations, and blue lines show the correlations between the modelled prior gradient and observations.**



For reference, the mixing ratio differences from two other global inverse models (CT-CH$_4$ and GELCA-CH$_4$) are shown in Fig. S9. Comparing the east-west differences, the posterior modelled $\Delta C_{ETL-FRD}$ from this study agrees better with the observed $\Delta C_{ETL-FRD}$. The results from CT-CH$_4$ and GELCA-CH$_4$ have poorer agreement with the observations (Fig. S9). Note that all three models used the observations from ETL and FSD (as well as EGB) to constrain their posterior fluxes. Yet they can perform differently when compared to the observed $\Delta C_{ETL-FRD}$. One possible explanation of the difference is that this study is a regional inversion focused on Canada, while CT-CH$_4$ and GELCA-CH$_4$ are global inverse models. The global model results are forced to minimize the global flux and mixing ratio errors as prescribed by the global cost function. In contrast, the regional model used here is mainly focused on minimizing the flux and mixing ratio errors for Canada (the regional cost function is not explicitly influenced by the flux and mixing ratio errors elsewhere). Similar results are seen in the correlation coefficients plots. This study has monthly correlation coefficients closer to unity after the inversions. The correlation coefficients in the global models could reach negative values and have little improvement after the inversions (Fig. S9). There appears to be an advantage for regional inversion compared to global inversion for regional flux estimates.

For the posterior north-south differences $\Delta C_{EGB-FSD}$ (approximately perpendicular to the prevailing westerly wind direction and less representative of the upwind downwind setup), all three inverse models (CT-CH$_4$, GELCA-CH$_4$ and this study) perform similarly when compared to observations (in Fig. S9), while the monthly correlations of this study are still better and more uniform with time than the other models (with lower correlations in general and more month-to-month variability). The global model results appear better in the north-south case compared to the east-west case. The fact that model results are closer to each other in their north-south differences (compared to the east-west) perhaps is an indication that the flux composition is distinct in the north-south arrangement (natural sources for the northern site (FSD) and anthropogenic sources for the southern site (EGB)). In contrast, West has a complex mix of anthropogenic and natural sources, compared to East where natural sources dominate; this source mixture appears to pose a bigger challenge to the global models. These differences are under investigation.

Overall, the mixing ratio differences over these larger spatial scales are useful as evaluation or verification data for the model results. Our regional inverse model shows better agreement with observed mixing ratio differences than the global inverse models examined. Some of the issues could be due to the differences between global and regional cost functions, the mixture of fluxes in each basis region, the amount of observations for the global inversions. More work remains to understand the differences among the models tested here.



## 3.4 Temporal variations of the fluxes

As presented in the previous sections, Sect. 3.1 and Sect. 3.2, the subregions North and West are not well constrained prior to 2012, due to limited observations in North. Notably, with larger observational data sets for the later period, 2012–2017, the inversion results are overall robust. Therefore, in the following sections, the results and discussion are focused on the temporal and spatial variations of the four subregional posterior fluxes from the reference inversion (Inv_4R12S) results, for 2012–2017.

### 3.4.1 Seasonal cycle

Figure 9 shows the ensemble mean comparison of the seasonal cycles between the posterior fluxes and the prior fluxes from 2012 to 2017 for each subregion. The prior fluxes in the subregions with significant wetland emissions (North, East and West) show strong maxima in the summer when wetland emissions are most active. The four different wetland priors have very different maxima, giving the large ranges of summer fluxes in Fig. 9. The posterior fluxes from inversions are much reduced in the summer, particularly in the East subregion containing the HBL. Differences in seasonal phase between the prior and posterior are evident in East and West. The spring increase in $CH_4$ emissions is delayed by about two months in the posterior fluxes compared to the prior fluxes, and the summer maxima appear late by one month from June–July to July–August. In September, the posterior fluxes remain high, compared to the priors. This seasonal phase shift in posterior fluxes suggests that surface air temperature is not the sole driver of the seasonality of wetland $CH_4$ fluxes. However, this result is consistent with the hysteretic temperature sensitivity of wetland $CH_4$ fluxes demonstrated by Chang et al. (2020), yielding more $CH_4$ emissions later in the warm season. Instead of a single temperature dependency in wetland model parameterizations, Chang et al. (2020) proposed the microbial substrate-mediated $CH_4$ production hysteresis; higher substrate (i.e., acetate and hydrogen) availability during the later period stimulates higher methanogen biomass. While Chang et al. (2020) validated their wetland model results with chamber field measurements, the present study examined the temperature dependency on a subregional scale, using our mean posterior fluxes and surface air temperature at 2 m above ground from NCEP reanalysis (Kalnay et al., 1996). As seen in Fig. 10, the posterior fluxes, especially in East and West, show that surface air temperature dependency varies between the early and the later periods in the warm season (air temperature > 0˚C), supporting the hysteretic temperature sensitivity hypothesis on the regional scale. The consistency between our results and Chang et al. (2020) is supporting the hypothesis that the wetland $CH_4$ emissions drive the seasonality of our posterior fluxes.





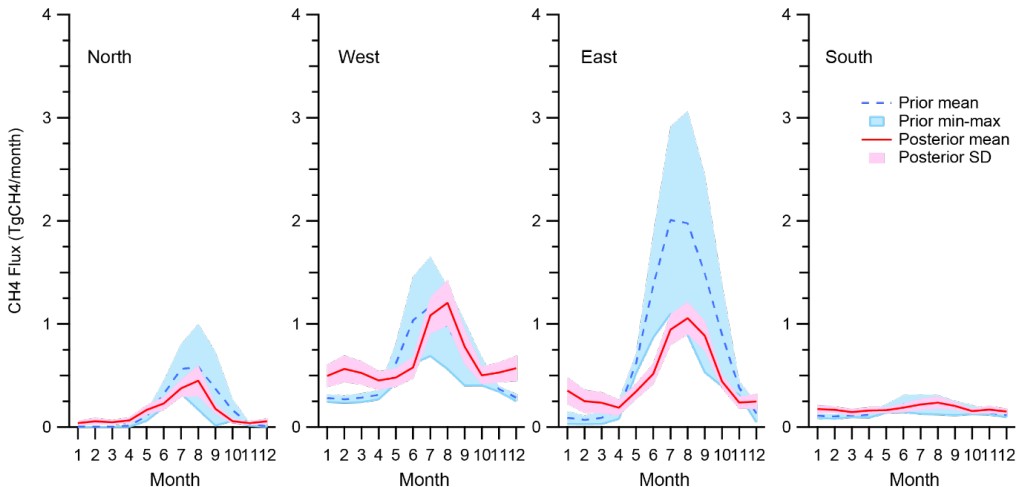

**Figure 9. Seasonal cycles of CH$_4$ fluxes for subregions (North, West, East and South). Red solid lines and red shaded areas indicate the mean posterior monthly emissions and standard deviations (SD) from 24 experiments in Inv_4R12S for 2012–2017. Blue dotted lines and blue shaded areas indicate mean prior emissions and their minima and maxima.**

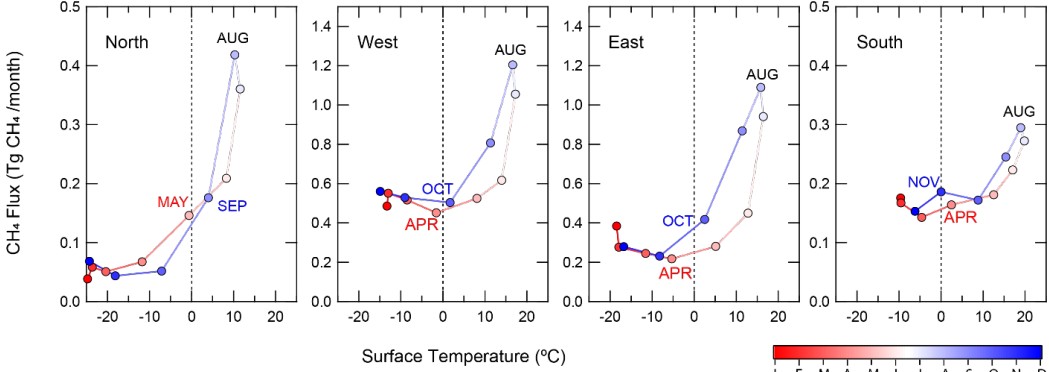

**Figure 10. Dependency of mean posterior monthly CH$_4$ fluxes on mean monthly air surface temperature. The inversion results and the air temperature data at 2 m above ground from NCEP reanalysis are averaged for 2012–2017 and aggregated over the subregions.**

In contrast to the reduced posterior summer fluxes, the posterior fluxes are higher during the cold winter season than the prior fluxes in both East and West. The presence of higher fluxes in East with little anthropogenic fluxes suggests the wetland emission in the winter is higher than the ecosystem model results used as priors. Also, our winter flux results are not consistent with the previous regional inversion results (e.g., Miller et al., 2014; Thompson et al., 2017), which do not show any large CH$_4$





emissions in the HBL in the cold season. The potential natural/wetland $CH_4$ emissions in the cold season is discussed in Sect.3.6. As seen in Figs. 9 and S10, the range of variation of fluxes for the prior is different from the posterior. As the dominant fluxes are from wetlands in the summer, the wide range of prior fluxes reflects the uncertainties in the different wetland process

models, including wetland types and spatial distributions, functional dependence on climate forcing, etc. Using the atmospheric $CH_4$ to constrain the summer emissions yielded posterior variations smaller than the priors. The inversion of the spatially distributed wetland fluxes (as seen in Figs. 3a and 3c) appears less sensitive to errors/differences among our transport models or prior flux magnitudes, giving smaller variations or more robust flux estimates in the summer posterior fluxes. In contrast, with locally non-homogeneous anthropogenic fluxes (Figs. 3b and 3d), transport errors appear more important in the winter,

resulting in larger variability in the posterior fluxes than the prior fluxes. The anthropogenic fluxes are predominantly distributed in western Canada; the posterior fluxes in West exhibit large transport dependency in the winter (Fig. S10).

### 3.4.2 Inter-annual variation and relationship with climate anomalies

As presented in Sect. 3.2, the inter-annual variations in the subregions among the inversions are comparable or greater than

the respective long-term trends. To examine the drivers of inter-annual variability of $CH_4$ fluxes, we focus on the later period from 2012 to 2017, when the inversions are constrained with better observation coverage. The inter-annual variability in the later period tends to be smaller than in the early period (Fig. 7). This tendency may be related to the amount of observational constraints on the inversions. The year-to-year change of posterior annual fluxes for Canada is relatively small; standard deviation (SD) of 0.6 Tg $CH_4$ year$^{-1}$ is ~3 % of the mean flux of 17.4 Tg $CH_4$ year$^{-1}$ (Fig. 7). In all the regions except for North,

the SD of the posterior annual fluxes exhibit < ~8%, while North shows ~18%, SD of the posterior annual fluxes (Fig. S6). The correlations of the posterior fluxes among the subregions and between the subregions and the nation are summarized in Table S3. Overall, no clear relationship among the subregional emission changes are found. There are no significant correlations of the year-to-year change in the posterior fluxes among the subregions, $r = -0.38$ to $0.37$ ($p > 0.43$), except for the correlation between East and South ($r = 0.82$, $p = 0.05$). The subregional flux changes are not correlated with the national

flux, $r = -0.12$ to $0.40$ ($p > 0.42$), while only North shows apparent correlation ($r = 0.74$, $p = 0.09$). Given the large geographical size of Canada, the low correlations of the posterior fluxes among the subregions indicate that the drivers of inter-annual variations of $CH_4$ flux may be subregional scale processes, such as synoptic systems (of temperature and precipitation variations) with weekly timescale and ~1000 km spatial scale. Furthermore, on a subregional scale, the summer flux changes appear to drive the inter-annual variations in subregional fluxes. The year-to-year change of the annual flux for each subregion

is well correlated with the summertime (July and August for North, July to September for the other subregions) flux anomaly within the respective subregions, $r = 0.97$ (North), $0.71$ (West), $0.90$ (East), and $0.77$ (South). The high correlations support that the change of natural summer $CH_4$ emission is a major factor in the inter-annual variability in Canada's $CH_4$ fluxes.



Thus, we examined the statistical correlation between flux anomalies and temperature anomalies by subregion, for the period
of 2012 to 2017.   For this, surface air temperature anomalies from NCEP reanalysis (Kalnay et al., 1996) are aggregated to
the respective subregions.   The correlations between the monthly flux anomalies and monthly surface air temperature
anomalies for the summer months are shown in Fig. 11. The inter-annual variability of posterior $CH_4$ fluxes for subregions
North, West and East exhibit moderate positive correlation with the surface temperature anomaly, $r = 0.64$ ($p < 0.01$, North),
$r = 0.60$ ($p = 0.01$, West), $r = 0.60$ ($p = 0.01$, East), except $r = -0.06$ ($p = 0.81$, South), as visualized in Fig. 11.

In South, no robust correlation is found between the posterior fluxes and climate on seasonal and monthly scales (Figs. 11d).
This is consistent with the prior fluxes in the south subregion being mainly (annually constant) anthropogenic fluxes and a
small component of natural wetland fluxes. It also serves as a check on the annually constant assumption for the anthropogenic
fluxes. The relatively strong flux-temperature dependence in three of the four subregions (Fig. 11) suggests that wetland $CH_4$
emission could be enhanced with climate change and warming.

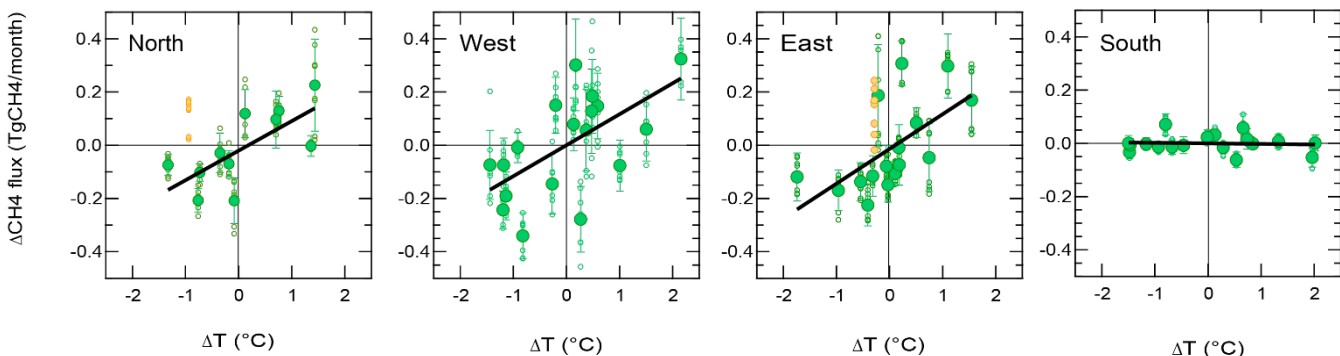

**Figure 11. Flux-temperature relationship in the summer season. Fluxes are the anomalies of estimated monthly fluxes from the 6-
year (2012–2017) mean monthly fluxes. Temperature is the regional monthly anomaly from the same 6-year (2012–2017) mean
monthly temperature. The summer season is defined as July and August for the North and July to September for the remaining
subregions. Closed green circles are the ensemble mean flux anomalies, error bars denote the SD, and green open circles are the
anomalies of 24 individual experiments. The yellow circles are from the estimated fluxes, which are excluded because the nearby
forest fires apparently affected the flux estimates.**

The correlation of inter-annual variability of posterior $CH_4$ fluxes with the surface temperature anomaly is evident on shorter
monthly timescale also. The correlations by month are shown in supplement, Table S4 and Fig. S11. Overall, the posterior
fluxes and temperature anomalies show positive correlations.  The wildfire component of the posterior flux, which is not
necessarily correlated to temperature, might affect the correlations between the posterior fluxes and temperature anomalies.
According to a fire monitoring system (Canadian Wildland Fire Information System, https://cwfis.cfs.nrcan.gc.ca), severe
wildfires started at the end of June 2014 around BCK in North, and continued until early August, causing higher level of $CH_4$





biomass burning emissions. Near FSD in East, in August 2017, local wildfire events in northwestern Ontario apparently caused high $CH_4$ emissions. If these possibly wildfire-induced positive $CH_4$ emission anomalies are removed, the positive correlations

with air temperature anomalies in North and East are improved, as monthly correlations are enhanced by ~50% (Table S4 and Fig. S11). Due to the short summer in North, clear correlations are found only for July and August. In West and East, the posterior fluxes in early summer, June, are less sensitive to air temperature anomalies than in the following summer months, including September. This high sensitivity summer period is consistent with the active summer natural emission period, as discussed in Sect. 3.4.1.

For comparison, the same analysis was done for the different prior fluxes with inter-annual variations (WetCHARTs, CLASSICdiag, CLASSICprog) used in this study, and the results are shown in Fig. S12. Only CLASSICdiag in East shows a positive temperature dependence ($r = 0.52$, similar to the inversion results in Fig. 11), the slope of the linear fit or flux-temperature sensitivity in CLASSICdiag is about half as large compared to the posterior flux. The other subregions and other prior fluxes show no clear dependence on temperature. These results suggest that many factors govern the wetland $CH_4$ fluxes.

For example, there are large differences in the spatial distribution for the different priors (see Fig.3a). More studies are needed to understand the flux-climate relationship better.

We also examined the correlation between flux anomalies with the precipitation anomalies with 0 to 2-month lag, but no significant correlation was found ($|r| < 0.2$, $p > 0.3$).

## 3.5 Spatial distribution of the fluxes

### 3.5.1 Total $CH_4$ emissions

Figure 12 shows the total (natural and anthropogenic) annual mean $CH_4$ emission estimates for Canada, compared with prior fluxes and previous inversion studies. Our mean posterior flux for Canada is 17.4 (range of min-max: 15.3–19.5) Tg $CH_4$ year[-1] and is near the lower end of the range of prior fluxes, but quite similar to the prior flux scenarios with CLASSIC prognostic wetland $CH_4$ fluxes (CLASSICprog_ECAQ and CLASSICprog_EDGAR). Compared with global inversion studies, our

estimate is slightly lower (by ~1.5 Tg $CH_4$ year[-1]) than GCP-$CH_4$ global inversion ensemble mean, but within uncertainties. However, CarbonTracker-$CH_4$ of 10.2 Tg $CH_4$ year[-1] (Bruhwiler et al., 2014) is substantially lower than our estimate. The regional inversions by Miller et al. (2014) and Thompson et al. (2017) do not cover the entire Canada, but cover partially south of 65˚ N and north of 50˚ N, respectively. Therefore, these differences should be noted in the comparison with these two previous regional inversions. The national flux estimate by Miller et al. (2014), 21.3 ± 1.6 Tg $CH_4$ year[-1] is more than double

of their priors (7.6–9.4). Miller et al. (2014) explained that the higher flux estimates might be attributed to the anthropogenic emissions in the province of Alberta in western Canada. Thompson et al. (2017) also estimated the larger anthropogenic emissions in Alberta, 4.3 ± 1.3 Tg $CH_4$ year[-1], nearly three time higher than their prior emission, EDGAR-4.2FT2010 (Janssens-



Maenhout et al., 2014). However, their estimated national total emission is slightly lower than our estimates, possibly because of their model domain. Thompson et al. (2017) did not include southern Ontario and Quebec, the densely populated area in
Canada.

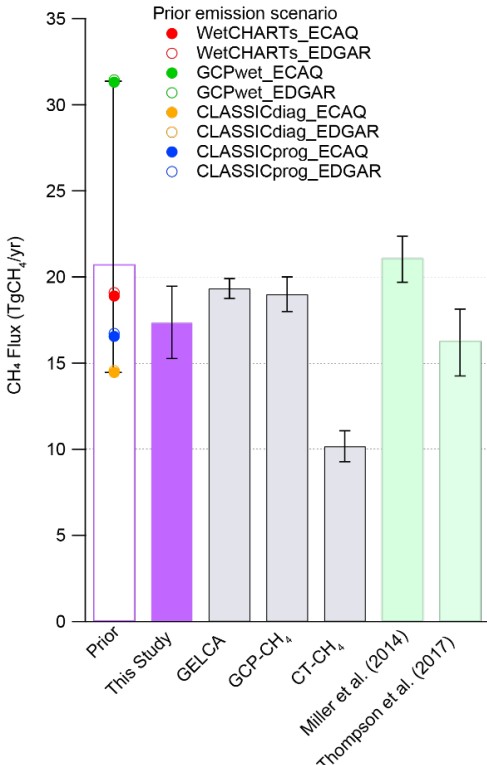

**Figure 12. Estimated mean total CH₄ emissions for Canada for 2012–2017 (purple solid bar). The mean of the eight prior flux scenarios is shown in the unshaded purple bar, along with each mean of individual prior scenarios. For comparison, the previous global and regional inversion results are plotted in gray and light green bars, respectively. Global: GELCA (2012–2017), CT-CH₄**
**(2000–2010) and GCP-CH₄ inversion ensemble (2008–2017). Regional: Miller et al. (2014) covers up to 65° N for 2007–2008, while Thompson et al. (2017) covers north of 50° N for 2005–2013.**

Some spatial differences can be seen in this study compared to the priors and global inverse models. At the subregional level, the mean prior flux distribution shows larger emissions in East than West (the spatial distributions of the prior scenarios are
shown in Fig. S13). Similarly, global inversions (CT-CH₄ and GELCA-CH₄) show larger emission in East than West. Only this study shows higher CH₄ emission in West than in East within Canada, (Fig. S13). For the subregion North, the Canadian Arctic, there are considerable differences among the priors, 1.4 to 4.2 Tg CH₄ year$^{-1}$. The CH₄ emission estimated in this study



is, $1.8 \pm 0.6$ Tg CH$_4$ year$^{-1}$, consistent with our previous study, $1.8 \pm 0.6$ Tg CH$_4$ year$^{-1}$ (Ishizawa et al., 2019, for the years 2012 to 2015).


**Figure 13. Mean spatial distribution of prior total flux (top), posterior total flux (middle) and the difference between posterior and prior (bottom). Partitioned posterior emissions into natural and anthropogenic sources by subregion, along with the respective prior emission means and ranges (min–max) are shown on the left and right sides.**




### 3.5.2 Natural and anthropogenic CH$_4$ sources

As a first attempt of total emission breakdown into natural and anthropogenic sources with the scheme presented in Sect. 2.3, we assumed $R_{cold}$ (fraction of cold season's natural CH$_4$ emission to its annual emission) to be in the range of 0 to 10% based on wetland models and previous studies (see Sect. 2.3 and Fig. 3e). Then, the anthropogenic emissions are approximately 12
Tg CH$_4$ year$^{-1}$ for Canada, 6.3 Tg CH$_4$ year$^{-1}$ for West and 2.8 Tg CH$_4$ year$^{-1}$ for East. These anthropogenic CH$_4$ emission values are much larger than the priors, more than twice as large as those for Canada and West from the prior inventories, and more than six times of the priors (~0.45 Tg CH$_4$ year$^{-1}$) for East.   These resultant national and subregional anthropogenic emissions seem excessive when compared to the several regional inversion studies with larger estimates of anthropogenic fluxes than the inventories, especially in West (e.g., Miller et al., 2014; Thompson et al., 2017; Chan et al., 2020; Baray et al.,
2021).   Even if there is potential for higher anthropogenic emission in West, the estimated anthropogenic emission in East seems unrealistic, where there is no significant anthropogenic CH$_4$ emitter according to the priors.

Next, we explored an alternative approach assuming that the natural CH$_4$ production is more active in the cold season than as predicted by the prior wetland models. Such a cold season wetland CH$_4$ emission has been reported by previous observation-based studies (e.g., Pelletier et al., 2007; Zona et al., 2016).  Zona et al. (2016) explained CH$_4$ emissions in Arctic tundra
continue even in cold season due to "the zero curtain".  When air temperatures drop to around 0˚C, there is a period when the water trapped in the soil below the surface has not freeze completely. Micro-organisms in the unfrozen layer remain active and emit CH$_4$ into the atmosphere. CH$_4$ emission in cold months, September to May, could account for $\geq$ 50 % of annual CH$_4$ emission in the Arctic.

As seen in Sect. 3.4.1, the posterior CH$_4$ fluxes in this study show notable winter emissions, which could be potential winter
(or cold season) natural/wetland fluxes in subregions with little anthropogenic CH$_4$, such as East and North (Figs. 9 and 10). Thus, we derived the winter natural flux fractions $R_{cold}$ from our estimated mean seasonal CH$_4$ fluxes, by assuming that the (seasonally non-varying) anthropogenic fluxes to be the mean prior anthropogenic fluxes of 0.45 Tg CH$_4$ year$^{-1}$ for November to March in East and 0.01 Tg CH$_4$ year$^{-1}$ for October to May in North (the uncertainties in these prior fluxes are examined below). Then, solving Eqs. (6) and (7), $R_{cold}$ is 22 (range of min-max: 20–24) % for East and 30 (29–32) % for North. We
applied the $R_{cold}$ derived for East to the West and South subregions, as these subregions are also located in the mid-latitudes with similar temperature/growing conditions.

The resultant mean natural and anthropogenic CH$_4$ fluxes for the subregions are shown in Fig. 13, along with the mean prior fluxes. For Canada, our estimate for natural emissions,  10.8 (range of min-max: 7.5–3.2) Tg CH$_4$ year$^{-1}$, is smaller than most of the process-based ecosystem model estimates, while our anthropogenic emission estimate, 6.6 (6.2–7.8) Tg CH$_4$ year$^{-1}$ is
larger than the inventories, 3.5 to 5.2 Tg CH$_4$ year$^{-1}$  (Ito, 2021; Stavert et al., 2021), primarily attributed to western Canada. The anthropogenic emission in this study for West, 5.0 (4.6–5.6) Tg CH$_4$ year$^{-1}$,  is comparable with previous regional





anthropogenic emission estimates (e.g., Miller et al., 2014, Thompson et al., 2017; Baray et al., 2021, Fujita et al., 2018). Chan et al. (2020) estimated nearly twice of $CH_4$ emission from the oil and gas sector in Alberta and the adjacent province Saskatchewan than NIR (higher by 1.6 Tg $CH_4$ year$^{-1}$), based on the 8-year wintertime atmospheric surface measurements.

Baray et al. (2021) also attributed their estimated higher national anthropogenic $CH_4$ emissions ($6.0 \pm 0.4$ Tg to $6.5 \pm 0.7$ Tg $CH_4$ year$^{-1}$) than NIR to western Canada ($4.7 \pm 0.6$ Tg $CH_4$ year$^{-1}$, for the provinces of British Columbia, Alberta, Saskatchewan and Manitoba), using ECCC surface measurements and Greenhouse Gases Observing Satellite (GOSAT) data to constrain their inverse model. Fujita et al. (2018) showed an additional time-invariant (anthropogenic) emission of $2.6 \pm 0.3$ Tg $CH_4$ year$^{-1}$ into the EDGAR inventory in Alberta to make their model simulation closer to the observed $CH_4$ mixing ratios at

Churchill.

One assumption in the flux partition analysis is that the posterior anthropogenic fluxes for the North and East subregions are the same as their priors in Eqs. (6) and (7). The sensitivity of the flux partitioning on this assumption was examined by repeating the analysis with halving and doubling their values. The results for the estimated anthropogenic $CH_4$ fluxes for Canada and the subregions West and South changed by < 5 %.  Thus, the flux partitioning appears stable and capable of detecting the

higher anthropogenic $CH_4$ flux from West.

Results for cold season natural $CH_4$ fluxes are wide ranging among recent studies, as cold season natural $CH_4$ fluxes are difficult to measure and quite variable in wetland model estimates. Treat et al (2018) reported measured cold (non-growing) season fraction of wetland $CH_4$, 16 % (95 % confidence interval CI, 11.0–23.0 %) between 40˚ N and 60˚ N, and 17 % (CI 16.0–23.3 %) for north of 60˚ N.  These fractions tend to be higher than process-based models (4–17 % within 40–60˚ N),

while the upscaled flux estimates based on the flux measurement with machine learning technique (Peltola et al., 2019) showed cold season emission (November to March) ~20 % for north of 45˚ N . Pelletier et al. (2007) reported up to 13 % of the annual emission in the winter (November to March), in peatland in James Bay Lowland, along the Hudson Bay coastline in Canada. A recently published $CH_4$ flux dataset from the flux measurement global network (FLUXNET-$CH_4$) has a considerable contribution of cold months (October to March) to annual $CH_4$ flux, $18.1 \pm 3.6$ % and $15.3 \pm 0.1$ % in northern (> 60˚ N) and

temperate (40°–60˚ N) regions, respectably (Delwiche et al., 2021).  An inter-comparison of 16 wetland models from the Global Carbon Project (Ito et al., 2023) showed cold season $CH_4$ fluxes (September to May) ranging from 11.6–40.1% in the Arctic (> 60˚ N), and 21.6–54 % north of 45˚ N. For comparison, the cold season (September to May) natural $CH_4$ emission in this study is 38.5 (38–39) % in the Arctic (> 60˚ N), and 51 (49–52) % north of 45˚ N. The natural $CH_4$ emission in this study is not directly comparable to the other wetland emissions as our natural $CH_4$ emission is limited to the model domain of

Canada and includes biomass burning, and soil sink. But our natural $CH_4$ emission estimate appears to be within the range of results of other studies. As the range of possible winter wetland emission fraction is large in previous studies, evidence of winter wetland/natural $CH_4$ emissions in our atmospheric $CH_4$ measurements is further examined in the following section.



**3.6 Signals of winter natural CH₄ emissions in observations**

The partition of total CH₄ flux into anthropogenic and natural components in Sect. 3.5.2 indicates the presence of natural
emission in the winter or cold (non-growing) season. In this section, we examine the observed atmospheric CH₄ to determine
if it is possible to see the signature of winter CH₄ emissions from the natural component. Emissions near an observation site
have a measurable temporal signal in the atmospheric CH₄. The atmospheric CH₄ is temporally modulated by the interaction
of the diurnal cycle in the Planetary Boundary Layer (PBL) with the nearby emissions, resulting in a diurnal cycle in the
observed atmospheric CH₄ with higher mixing ratio nighttime (during shallow PBL) and lower mixing ratio daytime (during
well-mixed deep PBL). This is the diurnal rectifier effect (Denning et al., 1996) for constant or slow-changing emissions. Other
factors like strong winds and cloudiness can affect the diurnal interaction. However, the coupling of PBL dynamics with local
emissions should be evident statistically in the monthly average diurnal cycle of the mixing ratios with sufficiently large
emissions (discussed below). Although the diurnal cycle of mixing ratio is a qualitative indicator of local emissions, it could
be viewed as a consistency evaluation of the inversion result.

The monthly mean diurnal amplitude at each measurement site was obtained as follows. Firstly, we calculated a normalized
diurnal cycle, defining the mean afternoon mixing ratio over the local times between 14:00 and 16:00 as a reference. Secondly,
the individual normalized diurnal cycles were averaged by month over the measurement periods (Fig. S14).  Then, we obtained
the monthly mean diurnal amplitude as a difference from the maximum mixing ratio for 24 hours, 0 UTC to 24 UTC. The
results are shown in Fig. 14, along with the monthly standard deviations (SD) of the afternoon atmospheric CH₄, which is a
measure of the variability of the (reference) afternoon mixing ratio. When the monthly diurnal cycle of atmospheric CH₄ is
larger than the monthly afternoon mixing ratio SD (a detectable signal above the expected variability), this is used as an indirect
indication of the presence of CH₄ fluxes around the measurement site.

For the baseline (coastal) sites ALT, ESP and WSA with negligible CH₄ fluxes nearby (Fig. 14), the diurnal cycles in
atmospheric CH₄ are absent as expected. For the urban site DWS located in the city of Toronto, with anthropogenic CH₄ fluxes
throughout the year, the diurnal cycle is much larger than the SD in both winter (59 ppb, averaging from November to March)
and summer (105 ppb, averaging from April to October). The summer diurnal cycle is much stronger than the winter as the
PBL has the strongest day-night contrast with strong solar heating during daytime and radiative cooling at nighttime. The
diurnal cycle for the near urban site EGB, ~100 km away from Toronto, is about 50 % as large as DWS with the strong
anthropogenic emission. Thus, the diurnal cycle of atmospheric CH₄ is sensitive to the emissions from an area of the order a
hundred-kilometre radius.

For the remaining arctic and boreal forest sites (INU, BCK, CHL, CHM, CPS, LLB, ETL, FSD), the strong summer diurnal
cycles of mixing ratios are clearly exhibited with amplitude of ~15 to 110 ppb (Fig. 14). Since these boreal forest sites are far
from anthropogenic sources, the strong diurnal cycles in summer are indications of summer natural CH₄ sources near the sites.





In the winter, there are clear diurnal cycles at LLB, ETL and FSD (5 to 20 ppb and clearly higher than the respective afternoon
SD). These sites are located near strong wetland sources according to the map of prior fluxes shown in Fig. 3a. Indication of
winter natural emissions in the observed diurnal cycle in mixing ratios is consistent with the inversion results discussed in
Sect. 3.5.2. The weaker winter diurnal cycles at INU, BCK, CHL, CHM and CPS suggest weaker natural emissions, which is
consistent with their locations of weaker prior fluxes (see Fig. 3a). Another factor for the weaker winter diurnal cycles in the
high latitudes or arctic region observation sites is the weak winter solar forcing and the correspondingly shallow PBL diurnal
cycles. Some caveats relating the diurnal cycle of mixing ratio at observation sites and inversion flux results are that the area
of influence for diurnal cycles in mixing ratio is much smaller than the inversion subregions.  Thus, though comparing an
indirect feature of emissions (diurnal amplitude) to the quantitative inverse flux estimates requires cautions, overall, the diurnal
cycles in observed mixing ratios appear consistent with the inversion results.

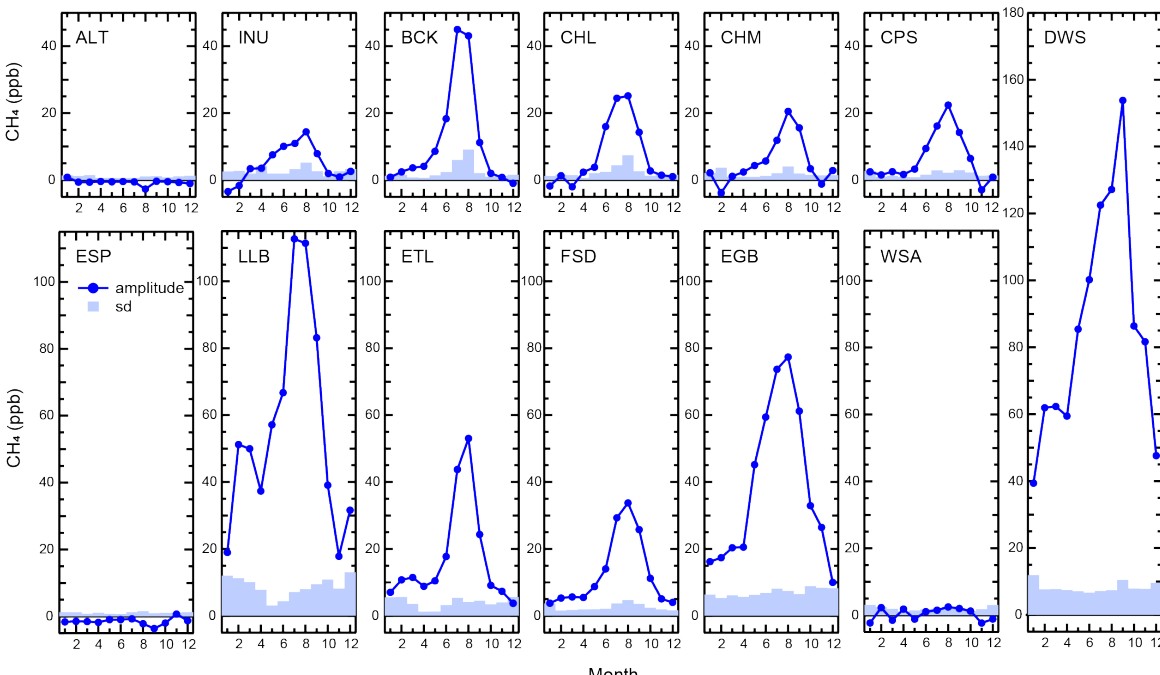


**Figure 14. Seasonal cycles of normalized diurnal amplitude and SD of observed atmospheric CH$_4$ during the afternoon mean (14–
16 local time base of normalization).**

## 4 Conclusion

In this study, we estimated the CH$_4$ fluxes in Canada using an inverse model constrained with ECCC's network of continuous
710  surface atmospheric CH$_4$ measurements. The Bayesian inverse model included an ensemble of prior fluxes and atmospheric
transport models to estimate the posterior flux uncertainties associated with these model variations. We analysed the variability



or robustness of the posterior flux estimates as a function of model resolution (number of subregions), quantity of observations (number of observation sites), as well as the spatiotemporal relationship of the posterior fluxes to climatological forcing.

Sensitivity experiments comparing different subregion masks (up to six subregions) were done to examine the variability of the posterior flux estimates. Results indicate that, with the set of 12 observation sites, the inverse model yields more stable and physical results (with no unphysical negative fluxes) for 4-subregion mask setting (Inv_4R12S, the reference inversion). The earlier period (2007–2011) with fewer measurement sites (without BCK, INU) has more variability in the flux estimates and unphysical negative posterior fluxes, primarily in western Canada.

The reference Inv_4R12S experiment ensemble mean estimate of total $CH_4$ flux for Canada (2012–2017) is 17.4 (range of min–max: 15.3–19.5) Tg $CH_4$ year$^{-1}$. This total is partitioned into 10.8 (7.5–13.2) Tg $CH_4$ year$^{-1}$ of natural sources and 6.6 (6.2–7.8) Tg $CH_4$ year$^{-1}$ of anthropogenic sources. In this study, the natural $CH_4$ source is still the major $CH_4$ emitter in Canada, especially in eastern Canada, though our estimated natural emissions are lower than most previous bottom-up estimates. By contrast, the anthropogenic $CH_4$ source estimates are higher than the inventory estimates, primarily in western Canada. The higher anthropogenic emission in western Canada is consistent with previous regional anthropogenic top-down emission estimates (e.g., Chan, et al., 2020).

The Inv_4R12S inversion results for 2012–2017 were analysed for other physical characteristics including the temporal, spatial and statistical properties as well as possible relationship to climatological forcing. Compared to other inversion studies, some notable results in our flux estimates include quantifiable amount of winter wetland $CH_4$ emissions (November–March with 20–24 % of the annual emissions for boreal regions and October–May with 29–32 % for sub-arctic region), hysteretic temperature dependence for wetland emissions over the warm (growing) season, and apparent correlation of wetland summer fluxes with mean summer temperature anomalies in Canada. No significant trend is found in the estimated decadal $CH_4$ emissions (the trend is within the flux estimate uncertainties), in contrast to some studies reported long-term trends. The differences from other inversion studies could be related to having more in situ measurement sites and focusing the inversion domain to Canada. This study also showed that the spatial mixing ratio gradients among the sites could be an independent verification tool of the posterior fluxes.

The measurement network across the nation is essential to improve our ability not only to quantify how Canada's natural $CH_4$ emissions will respond to climate change, but also to monitor anthropogenic $CH_4$ emission trend in response to Canada's $CH_4$ reduction efforts. These regional inversion results, which reflect better observational constraints on regional scale of space and seasonal time scale, might help wetland process models to improve their sensitivity and functions to climate parameters. Improvements in wetland process models and anthropogenic emissions could lead to improving regional climate model predication.



*Data availability.* The ECCC observations are available at World Data Center for Greenhouse Gases (WDCGG, https://gaw.kishou.go.jp) and the NOAA Observation Package website (ObsPack, https://www.gml.noaa.gov/ccgg/obspack).

*Author contributions.* MI and DC designed the research. MI conducted all inversions and data analysis, wrote the initial draft, and edited together with DC. DW led the ECCC GHG measurement program. MI, DC, and EC provided and processed footprint data for the inversion. JM and VA provided the CLASSIC wetland $CH_4$ fluxes. All co-authors reviewed the manuscript and contributed to the discussion and revision.

*Competing interests.* The authors declare that they have no conflict of interest.

*Acknowledgements.* We are grateful to the NOAA CarbonTracker Lagrange (CT-L) program for providing the WRF-STILT footprint data of Canadian sites for our inversion study. We acknowledge the Global Carbon Project $CH_4$ modelling group (Saunois et al., 2020) for the prior wetland $CH_4$ emission data used in this study and the posterior national emission estimates for Canada, and Lori Bruhwiler for providing CarbonTracker-$CH_4$ results and Rona Thompson for providing FLEXINVERT $CH_4$ inversion results for the analysis in this study. We thank the team members of ECCC's Greenhouse Gases Measurement
Laboratory, Larry Giroux Bob Kessler and Senen Racki to collect the observations and maintain the ECCC GHG measurement network.

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
