# Peer review of "Estimation of Canada's methane emissions: inverse modelling analysis using the ECCC measurement network"

_EGUsphere, 2023_

## Author Comment (AC1)

**Reply to Comments by Referee #1**

We are thankful for the referee's constructive comments and suggestions, which helped clarify and improve our manuscript. Below are our responses, copying the comments *in italic and red*. In the responses, we also indicate the changes made in the manuscript in blue.

**General comments**

*Ishizawa et al. conduct inverse modeling simulations for Canada to estimate wetland fluxes across the country. I think this manuscript addresses several very important topics and makes an important contribution to the literature on methane fluxes from high latitudes. I do worry about some aspects of the methodology, and I think there are more state-of-the-art approaches to inverse modeling that would mitigate some of the unrealistic or unphysical results that the authors highlight at several points in the manuscript.*

We appreciate the referee's comment on our contribution on methane flux estimation from high latitudes. The general and specific comments raised some questions on the selection of inversion model used in this study. Notably that we didn't use non-negative flux constraint to eliminate negative fluxes and high resolution (grid point) inversion model to better resolve fluxes. We have included below more detailed explanations and evaluations (pros and cons) of the different inversion modelling approaches and the reasons for the selection of the model used in this study.

The Bayesian inversion model used in this study is a statistical (non-deterministic) optimisation technique for our linear mixing model:

**C = MS + E,**

where **C** = concentration, **M** = atmospheric transport (linear operator), and **S** = sources (emissions), and **E** = errors.

In our model testing, the Bayesian inversion model worked well if the basic model assumptions are satisfied. The important assumptions are (1) no transport errors, and (2) a large data set for robust statistics. From the model testing, we found that the main reason for the negative posterior fluxes is model transport errors (the inversion model yields the best statistical fit of the observations without accounting for transport biases). By comparing multiple transport models, we found that the largest source of inversion model errors in the flux estimates is transport model bias errors.

For atmospheric transport with random errors (unbiased), the model still works well if there are sufficient constraining data ('observations') to allow the statistical model to robustly estimate the scaling factors. Imposing positive flux constraints (usually for negative solutions resulting from scarcity of constraining data (e.g., Michalak and Kitanidis, 2003)) does not appear to be addressing the problem of transport biases. Positive flux constraints, or imposing nonnegativity constraints on the scaling factors, could violate the statistical assumptions in our linear Bayesian inverse model, namely linearity and normality.

As noted by the referee, there are studies doing grid scale inversions to address the aggregation errors issue (e.g., Gourdji et al., 2012; Hu et al., 2019; Thompson et al., 2017), but their discussions are typically on the aggregated fluxes to larger regions and temporally averaged estimated features. Gourdji et al. (2012) concluded that 'an expanded measurement network will further help to reduce the sensitivity of inversion results to setup assumptions, although systematic transport model errors will still remain a concern.' It seems that the additional covariance assumptions needed for grid point inversion could not account for the lack of observations and statistical data requirements. Grid inversions appear to have similar statistical and transport error limitations as other inversion models using larger sub-regions. This is consistent with our inversion model sensitivity analysis; we found that inversion flux errors from the transport model errors appear larger than aggregation errors in our case. For example, in the worst-case scenario, the difference in the inversion results calculated by different transport models could be greater than 100%.

In this study, we tried to reduce the effects of transport model biases and insufficient observations for statistical Bayesian inversion analysis. We used multiple transport models to lessen the effects of individual transport model biases and to provide flux uncertainty estimates associated with the choice of transport model. The inversion model was tested with different sub-region definitions (2, 4, 6 sub-regions) to obtain flux estimates that appear robust and positive without the added model complications like positivity constraints and non-zero covariance matrices. We were able to analyse the spatiotemporal characteristics of our flux estimates as reported in the Results and Discussion section.

Flux estimations by inversion models could be complex, as noted above, the flux estimate uncertainties depend on the tracer bio-geochemical characteristics, quantity and quality of the observations, model formulation, setup and assumptions. Having a wide range of models (including grid point inversion, non-negative constrained inversion, multi-transport with abundant observational constraint inversion used here, etc.) could be helpful to understand the strengths and weaknesses of inversion modelling. We have added more explanations in the revised manuscript to provide the motivations and reasons for our model setup in response to the specific referee's comments.

It is not clear what spatiotemporal information of fluxes could be retrieved (robustly and without significant or large negative fluxes for the $CH_4$ case, at the minimum) from a given set of observations using a Bayesian statistical optimisation model. We are investigating how to improve our inversion model to resolve fluxes better. Therefore, we need to understand better the potential improvements and limitations for our case (inferring $CH_4$ fluxes for Canada).

We have a study in progress to understand some of the critical factors governing the spatiotemporal resolution of the fluxes that could be resolved, using model-simulated $CH_4$ concentrations as the 'observations' (pseudo-observations study). Some of the factors we are examining include: how much observation and their signal contents or strength (synoptic variability for our case and the amount of flux signal mixing as the air flowed over different sources),

seasonal variations in transport (winds and PBL characteristics), transport model biases (including background or boundary conditions uncertainties and biases, the impact of additional transport models, on inversion results), spatial and temporal variations in fluxes and various settings used in the inversion model. General statements on how well inversion models can resolve posterior fluxes spatiotemporally are difficult. We have some modelling results indicating that due to the diffusive nature of atmospheric transport and the large samples of observations required, the inversion model appears to have limited ability to resolve posterior fluxes for the flux types, model domain and observation network in our study, and may not be able to resolve grid scale fluxes. We will report our findings and the possible improvements for the inversion model after our series of model evaluations and sensitivity studies.

**Specific comments and suggestions:**

1. *Line 15: What kind of gradients are you referring to here? Spatial gradients, vertical gradients, or some other kind of gradient?*

We mean the differences in $CH_4$ mixing ratios among the sites by gradients. We have modified to clarify as below:

First, the modelled  differences in atmospheric $CH_4$ among the sites show improvement after inversion when compared to observations, implying the $CH_4$  observation differences could help verify the inversion results.

2. *Line 28: What commitment are you referring to here? Does this line refer to thee Global Methane Pledge or to some other commitment?*

Yes, we are referring to the Global Methane Pledge as giving the reference, CCAC (2023). We have specified it as below:

the Global Methane Pledge has been launched at COP26, as a global effort to reduce anthropogenic $CH_4$ emissions by at least 30 percent from 2020 levels by 2030 for global climate mitigation (CCAC, 2023).

3. *Line 52: I'd recommend adding a transition at the beginning of this paragraph. Otherwise, the pivot from wetlands to natural gas feels really sudden and abrupt.*

We started the paragraph, adding a sentence at the beginning as below:

Canada's anthropogenic emission rates also have differences between inventories and measurement-based estimates. Canada is the fourth largest producer of natural gas, ..

4. *Line 62: This line mentions that reliable anthropogenic emissions estimates are important for regulation. Are there relevant regulations on methane emissions in Canada?*

Yes, there are regulations on methane emissions by federal and provincial governments in Canada. We modified the line adding the following phrase and references:

to regulate Canada's anthropogenic $CH_4$ emissions as set by both federal and provincial governments (e.g., Government of Canada, 2018; 2023; Government of Alberta, 2018; Government of Saskatchewan, 2019).

5. *Line 112: The phrase "minimize the impact of local sources" sounds a bit ambiguous or confusing here. For example, it wasn't totally clear to me what "local sources" means in this context or why those sources are bad. Instead of this motivation, I think a stronger motivation is that nighttime and morning mixed layer dynamics are really tricky to simulate, and atmospheric transport models don't always do well at this task. This difficulty can lead to large errors in the atmospheric transport model, which can potentially interfere with the inverse model.*

As the referee noted, the model simulation is challenging when the PBL is not well mixed, such as, nighttime, and early morning. In this study, we used the afternoon mean atmospheric $CH_4$ measurements, which are assumed to represent the well-mixed $CH_4$ sources within a region, as aiming to estimate regional-scale emission (~100–1000 km). Occasionally, the weather condition and local source ($<< \sim100$ km) interact with each other to yield large $CH_4$ mixing ratio. These high mixing ratio events are difficult to capture in transport model. The large model-observation mismatches could lead to large flux estimate errors. That is why we minimize the local cause by using the mid-afternoon mixing ratios representative of larger regions.

We rephased local sources with local-scale variations of atmospheric $CH_4$.

6. *Line 115: What percentage of data are removed as outliers? What do you think causes these outliers (i.e., what are the possible "unknown sources" cited in line 114)? I can see the rationale for removing outlier data, but I also think it's important to ensure this step doesn't eliminate the atmospheric signal from important emissions sources.*

Less than 2 % of the data were removed as outliers. Even during the daytime, local (point) sources can interact with local-scale atmospheric condition, leading abrupt or sporadic spikes in observed atmospheric $CH_4$. The $CH_4$ emissions from oil and gas sectors are mainly fugitive, which could emit irregularly from the ventilation during the extraction or refinery processes or storage tanks.

We also flag the observations if there are any issues with air sampling or instruments. We assume that regional-scale emissions are reflected on synoptic-scale variations of the observations. For clarification, we rephrased "unknown sources" as follow:

which indicate contamination  probably from sporadic strong local fugitive emissions, issues related to air sampling or analysis.    Less than 2 % of the data were removed through this process.

> 7.  *Line 136: What is the temporal frequency or resolution of these scaling factors? I.e., do you estimate a single scaling factor in each region and apply it to the entire study period (years 2007 - 2017), or do these scaling factors vary by month/season/year? The way that lambda is defined in line 136 seems to imply that there is only a single scaling factor in each region for the duration of the inverse model.*

The temporal resolution of scaling factor is monthly as we state at the end of this section, Sect. 2.2.1. We estimated scaling factors for each month per region for the period of 2007–2017, that is, totally 12×11 per region. For simplicity, we describe how the scaling factors are optimized for one time step, namely one month in our study.  We have modified the preceding sentence as follows:

N is the number of  observation points times number of stations (N is for one month in our case and is reduced if observations are missing). $\lambda$ (R×1) is the vector of the posterior scaling factors.

> 8.  *Line 144 and 145: What are the units on these sigma values, and why were these particular values chosen?*

They are unitless, which could also be expressed in percentage. The sigma value for prior uncertainty, $\sigma_{prior} = 0.30$ (or 30 %), is from the uncertainty in the $CH_4$ emission used in Zhao et al. (2009).  The prior model-mismatch, $\sigma_e = 0.33$ (or 33 %), is comparable to those used in previous regional inversion studies (e.g., Gerbig et al., 2013; Lin et al., 2004, Zhao et al., 2009), which considered different error components such as LPDM dispersion, wind field, aggregation, and background $CH_4$ mixing ratio. This estimate is based on many assumptions that are difficult to evaluate. In our previous study (Ishizawa, et al. 2019), we have tested the sensitivity of the inversion results to various setting by using 0.33 and 0.66, as the model-data mismatch errors, $\sigma_e$. The posterior fluxes changed by less than 5% for all sub-regions,

indicating that the flux estimates were not highly sensitive to the prior error specification. Therefore, we decided to use $\sigma_e = 0.33$. We have added the references for these sigma values in the manuscript and modified the text as follows:

We assigned $\sigma_e = 0.33$ (uncertainty of 33 %) for the model–observation error (Gerbig et al., 2013; Lin et al., 2004; Zhao et al., 2009) and $\sigma_{prior} = 0.30$ (uncertainty of 30 %) for the prior uncertainty (Ishizawa et al., 2019) (Zhao et al., 2009), as examined in Ishizawa et al. (2019).

9. *Line 167: Is 5 days sufficient for the back trajectories? I think John Lin and Christoph Gerbig used 10-day back trajectories in their original studies of $CO_2$ fluxes from North America, and many regional inverse modeling studies for North America use 10-day back trajectories (e.g., existing studies using CarbonTracker-Lagrange footprints, including those from Sharon Gourdji and Yoichi Shiga). Is 5 days sufficient time for the modeled particles to reach the edge of the modeling domain?*

Particle travelling time and size of the domain of the interest are related. Bigger space/domain need more time (for the air particles to travel over). As seen in Figure S4, the 5-day footprint covers the Canadian land, the domain of our interest and the particles after five days are mostly outside Canada. Furthermore, most synoptic-scale variations in the atmospheric mixing ratios at measurement sites are sufficiently explained by footprints within two to five days after particles are released (see Fig. R1a). Previous studies (e.g., Cooper et al., 2010; Gloor et al., 2001; Stohl et al., 2009) have shown that five days are typically sufficient to capture the surface influence on a measurement site from the surrounding region.

We also tested how the back trajectory duration would affect the flux estimation by expanding the duration to 10 days. Note that the footprint becomes more and more uncertain with longer and longer back trajectories. It is still being determined whether adding the footprint after five days is beneficial. The choice of the back trajectory duration slightly changed the modelled prior mixing ratio, comparable to ~10% of the uncertainty with the transport error from the different transport models used in this study (see Fig. R1b). The difference in the estimated fluxes between 5- and 10-day back trajectories is not significant, less than 5 % for regional fluxes. Thus, we concluded that 5-day back trajectories are sufficient in the scope of our study.

[Figure]

**Figure R1. (a) Example of the annual mean footprint contribution during 10-day back trajectory for East Trout Lake. The integrated footprint by day is normalized with respect to the total footprint for the entire period of 10 days. Footprint only over the Canada domain is included. Day back 0 is when the particles released at 14 LT until the first 0 UTC, and the following Days back are defined for every 24 hours. (b) the mean differences in the prior modelled mixing ratios between 5-day footprint and 10-day footprint with the standard deviation (blue), along with the mean transport uncertainties among three atmospheric models used in this study for respective sites (orange) for comparison.**

We have added the following text for clarification at the end of Sect. 2.2.2 and included the above Fig. R1 as Fig. S15 in the supplementary information.

There are many factors governing the transport model simulated concentrations at the measurement sites used in this study, such as the spatial distribution of emissions and meteorological conditions, including winds and atmospheric stability. Since we are focusing on the synoptic variability in our observations, these are the results of the regional emissions (typically within the synoptic spatial scale of ~100–1000 km). This region of interest is covered by the model footprint within the first three to five days. Another reason for limiting the footprints to five days is that footprint uncertainty grows the longer the hindcast or model dispersion (analogous to forecast uncertainty). Using 5-day footprints in the inversion model is similar to other studies (e.g., Cooper et al., 2010; Gloor et al., 2001; Stohl et al., 2009). These studies have shown that five days are typically sufficient to capture the surface influence on a measurement site from the surrounding region. Figure S15a shows the typical footprint strength as a function of days of hindcast. Figure S15b illustrates the differences in simulated concentrations between 5-day and 10-day footprints for our measurement sites and how they compare (~10%) with the much larger differences resulting from using different transport models in this study. Therefore, we used 5-day footprints in our model and included the footprint contribution beyond five days implicitly as a part of the background mixing ratio extracted at the 5-day particle endpoint locations. Other inversion studies could optimize emissions far from the observation sites (with weak nearby emissions); it would be necessary to consider footprints from five days to ten or more days in such cases, even though

the footprint (or model transport) uncertainties could become large (possibly) and lead to correspondingly large uncertainties in the inversion results.

10. *Line 189: What do you mean by "stronger emissions"? Is that the same as "larger emissions"?*

Both are similar relative expressions. To make them specific, we have revised as below:

Overall GCPwet shows stronger emission areas  among the four prior summer wetland fluxes, especially along Hudson Bay and around the border between Northern Territories and Alberta in western Canada, resulting in  approximately doubled annual emissions at subregional and national levels then the other estimates (Fig. 3c).

11. *Figure 5: Both the two and four-region inverse modeling setups seem relatively coarse. The ECCC network has 13 sites (i.e., Table 1). Personally, I think that using only two regions in a scaling factor inversion is really under-utilizing the ECCC network. By contrast, existing regional-scale atmospheric inverse modeling studies often estimate emissions at the model grid scale (e.g., see inverse modeling studies by Sharon Gourdji, Yoichi Shiga, Lei Hu, and Nina Randazzo). Furthermore, I'm worried that the relatively coarse regions used here might mean that the inverse model can't differentiate anthropogenic emissions in Alberta from wetland emissions in other parts of western Canada. As the authors point out, existing inventories tend to greatly underestimate emissions from oil and gas operations in Alberta, while some of the wetland models overestimate wetland methane emissions. The regions in the inverse model might be so coarse, that the resulting scaling factors won't differentiate these contrasting discrepancies between anthropogenic and wetland bottom-up emissions estimates in western Canada.*

The 2-subregion mask was introduced to examine the trend of posterior fluxes when there was no measurement in North before 2012. It is because, without sufficient observation coverage, the 4-subregion inversion produces unrealistic fluxes, as shown in Fig. 6.

As we mentioned in our response to the general comment, the grid inversion does not necessarily optimize the fluxes better with limited observations. Without proper evaluation of the posterior fluxes from grid inversions, it would be hard to determine the advantage at this point.

12. *Line 267: "in the North" instead of "in North"?*

We intend to use "In North" in the manuscript as "North" is the name of the region (similar to "in Canada" rather than "in the Canada"). In the 4-subregion mask case of this study, we named Canadian Arctic region, "North". Similarly, the three other subregions are named West, East and South.

13. *Line 298: There are ways of constraining the fluxes to be non-negative. For example, you could use a data transformation or a bounded optimization algorithm (Matlab and Python, for example, have several functions that do bounded optimization. These algorithms include active set minimization algorithms and LBFGS-B, among others.). I think there are also other possibilities for why the fluxes are negative here, including errors/uncertainty in background methane levels.*

We appreciate the referee's suggestion of using an optimisation algorithm. As we explained in our response to the general comment, we conducted this study without such optimization algorithm.

As commented by the referee, background levels could be a source of negative fluxes if they are higher than the mixing ratios at the measurement sites. However, this is not the case for the negative posterior fluxes in our 4-subregion inversion. As shown in Figs. 6 and S3, the negative fluxes are estimated for North when no observations available within the subregion of North.

14. *Line 300: This statement sounds like it belongs better in the methods section than the results.*

We moved the statement into the method section, Sect. 2.2.5, which has been revised as follows:

**2.2.5 Experimental setup**

Figure 5 shows the schematic diagram of the inversion experiments regarding the combinations of prior fluxes, transport models, subregion masks and observations. The ensemble of 24 experiments consists of the permutations of eight prior flux scenarios and three transport models, as summarized in Table S1.

It is noted that the number of the maximum observation sites was 12 in this inversion, as Alert (ALT) was not used for the flux estimation. The marine boundary layer site, ALT at the northern end of the subregion North, appears not to see the subregional flux signals (mainly in the southern part of the subregion) above the background atmospheric $CH_4$ (Ishizawa et al., 2019). Therefore, ALT was not included in the inversion of this study, following the inversion study of Canadian Arctic $CH_4$ (Ishizawa et al., 2019). As the reference inversion, we performed these 24 experiments with the 4-subregion mask and all 12 site observations (abbreviated as Inv_4R12S). As a sensitivity test to examine the impact of on observational coverage, two additional inversions using the 2-subregion mask with the 12 sites

(Inv_2R12S) and two sites of ETL and FSD (Inv_2R2S) were conducted with the same ensemble setup of 24 experiments. ETL and FSD have long measurement records extending back beyond the period of this study. Therefore, the inversion Inv_2R2S explored the feasibility of estimating $CH_4$ fluxes by inversion for a longer time period.

15. *Line 321: You definitely wouldn't want a reader to misinterpret these statements in the manuscript and think the inverse model is faulty or untrustworthy (and that your results are therefore untrustworthy). Another possibility here is to look at the posterior uncertainties. Presumably, the posterior uncertainties are large in years when there are few observations. The posterior best estimate might be unrealistic, but the uncertainty bounds could very well encompass realistic values. Overall, I think the "Does it make sense?" litmus test is one way to evaluate the uncertainties in the posterior flux estimate, but the posterior uncertainties are another way to do that. And again, enforcing non-negativity within the inverse model (see above) would be another way to eliminate these unrealistic, negative flux estimates.*

The statements on 'physically realistic solutions' could be misinterpreted as the referee's comment noted. We have revised to clarify the text on the reduction in the posterior uncertainties as follows:

The subregion West (on the south side of the subregion North, see Fig. 4a) also shows more variability in the posterior fluxes before 2012, particularly in the 2008 and 2010 winters. The presence of the poorly constrained North before 2012 (an extra degree of freedom in the inversion) appears to influence the statistical optimization of the inverse model as a whole, leading to more temporal variability and larger posterior uncertainties in the posterior fluxes in West. As noted from 2012 onward, there appear to be sufficient sites and observations to constrain North. Consequently, the posterior fluxes for West also show less variability and a reduction in the posterior uncertainties or more robustness after 2012.

16. *Line 330: What kind of variability are you referring to in this line? Also, see the comments above about how to add a non-negativity bound to the inverse model. Again, you wouldn't want a reader to think that the inverse model is untrustworthy.*

We have clarified that the 'variability' is 'temporal variability' and used the better description 'large uncertainties' for the posterior $CH_4$ fluxes. The revised text is as below:

In Sect. 3.1, the inversion results with four subregions and 12 observation sites (reference inversion, Inv_4R12S) show large temporal variability  and uncertainties for the posterior $CH_4$ fluxes for some subregions in the early period (2007–2011), compared to the later period.

*17. Lines 330 - 340: I don't think that the use of fewer subregions is the solution here. Lots of people in the inverse modeling community have estimated $CO_2$ and $CH_4$ fluxes across North America and have estimated those fluxes at the model grid scale (i.e., see the list of authors in an earlier comment). Rather, I think that enforcing non-negativity in the inverse model is a much better path forward. In addition, the purpose of the prior covariance matrix is to regularize problems that are under-constrained by the data. For example, you can include off-diagonal elements in the prior covariance matrix, and these terms will push the inverse model to estimate scaling factors that a correlated from region-to-region (i.e., correlated spatially) or correlated in time. In summary, an advantage of using a Bayesian approach to inverse modeling is that it can accommodate problems that are under-constrained, and I recommend taking advantage of those aspects of the Bayesian approach.*

We agree that Bayesian inversion models employ many different assumptions to obtain flux estimates. Each approach has advantages and disadvantages, and it would be useful to have these different inversion models (including the model used in this study) to help understand the strengths and weaknesses of the different inversion models. As explained above, the inversion model in this study included different transport models to help account for transport biases and used the wealth of observations to provide more robust flux estimates for Canada. The posterior fluxes from this study have uncertainty estimates that included errors from the three transport models, the eight sets of prior fluxes and different model setups. A comparison with other studies shows good general agreements, but also interesting new spatiotemporal features and relationships to climatological forcings. We hope this study has contributed to the science of inversion modelling and helped provide more insights into the $CH_4$ cycle in Canada. In the revised manuscript, we have added the following explanations on the inversion model differences and their pros and cons at the end of Sect. 2.2.4 Domain and subregions.

In our model testing, the statistical Bayesian inversion model worked well if the basic model assumptions are satisfied. The important assumptions are (1) no transport errors and (2) a large data set for robust statistics. We found that the main reason for the negative posterior fluxes in our model is transport errors (the inversion model yields the best statistical fit of the observations without accounting for transport biases).

For atmospheric transport with random errors (unbiased), the model still works well if there are sufficient constraining data ('observations') to allow the statistical model to robustly estimate the scaling factors. Imposing positive flux constraints (usually for negative solutions resulting from a scarcity of constraining data (e.g., Michalak and Kitanidis, 2003)) does not appear to be addressing the problem of transport biases. Positive flux constraints, or imposing non-negativity constraints on the scaling factors, could violate the statistical assumptions in our linear Bayesian inverse model, namely linearity and normality.

There are inversion studies doing grid scale inversions (using non-zero off-diagonal covariance constraints) to address the aggregation errors issue (e.g., Gourdji et al., 2012; Hu et al., 2019; Thompson et al., 2017). These grid scale inversions are limited by the lack of observations and systematic transport errors (Gourdji et al., 2012). Their discussions are typically on the aggregated fluxes to larger regions and temporally averaged estimated features. This is consistent with our inversion model sensitivity analysis; we found that inversion flux errors from the transport model errors appear larger than aggregation errors in our case. For example, in the worst-case scenario, the difference of the inversion results calculated by different transport models could be greater than 100%.

In this study, we tried to reduce the effects of transport model biases and insufficient observations for statistical Bayesian inversion analysis. We used multiple transport models to lessen the effects of individual transport model biases and to provide flux uncertainty estimates associated with the choice of transport model. This inversion model employed a limited number of sub-regions to allow the abundant observations to provide sufficient constraint to obtain flux estimates that appear robust and positive without the added model complications like positivity constraints and non-zero off-diagonal covariance constraints.

Flux estimations by inversion models could be complex; the flux estimate uncertainties depend on the tracer bio-geochemical characteristics, quantity and quality of the observations, model formulation, setup and assumptions. Having a wide range of models (including grid point inversion, non-negative constrained inversion, multi-transport with abundant observational constraint inversion used here, etc.) could be helpful to understand the strengths and weaknesses of inversion modelling.

18. *Lines 355-357: Presumably, one could answer this question by looking at the posterior uncertainties.*

East_2 in Inv_2R2S shows the larger posterior uncertainty ($\sigma = 0.08$ Tg year$^{-1}$) than other two inversion cases ($\sigma = 0.06$ Tg year$^{-1}$). As well as the posterior uncertainties, we point out that insufficient observational coverage would bias the trend.

19. *Line 378: Maybe "estimate" should be "estimates"?*

Corrected

20. *Line 393: What do you mean by "assimilated well"? Can you use a different phrase here to clarify the meaning?*

We have changed from 'assimilated well by the background mixing ratios' to 'simulated well by the background mixing ratios.'

21. *Line 465: If not solely temperature, what other drivers do you think are key?*

The drivers of wetland $CH_4$ seasonality are complex and remain as knowledge gaps in wetland $CH_4$ models and flux measurement, though temperature has been widely found a strong driver to constrain wetland $CH_4$ emission (Delwiche et al., 2021). Carbon substrates and inundation/water table would be also drivers and predictors of wetland $CH_4$ seasonality. Based on flux measurements along with a biogeochemistry model, Chang et al. (2020) demonstrated that the seasonal $CH_4$ emission is not a simple-single valued function of air temperature, but modulated hysterically by the availability of microbial substrates.

We have added the following sentences:

Temperature has been widely found a major driver to constrain the seasonal cycle of wetland $CH_4$ emissions, but their relationship might not be linear. Other factors, such as carbon substrates and seasonal inundation, also driver wetland $CH_4$ seasonality (Delwiche et al., 2021).

22. *Line 489: I think this statement represents the challenge of using such large regions in the inverse model. There are some cities in the Eastern region. If one used smaller regions or did a grid-scale inversion, then it would be easier to zoom in on wetland regions like the Hudson Bay Lowlands or the wetlands near Chapais, Quebec.*

In the 4-subregion mask of this study, the biggest city in East is Winnipeg, the capital of the province of Manitoba, while the major cities in eastern Canada, such as, Toronto in Ontario and Montreal in Quebec, are within the subregion of South in this study. As shown in Fig. 3d, the anthropogenic $CH_4$ emissions in East are ~0.04 Tg year$^{-1}$, mainly from the agriculture sector. Thus, we assumed the possible source of the winter $CH_4$ emissions would be natural. As mentioned in our reply to the General comment, we do not expect that a grid-scale inversion could help determine the source regions, given the spatial coverage of the available observations.

23. *Line 491: What do you define as large here?*

The winter fraction of CH$_4$ emissions in our study is 22 % for East as presented in Sect. 3.5.2 while those from previous regional inversion studies are around 10 % or less.

The manuscript has revised as follows:

Also, our winter flux results are not consistent with the previous regional inversion results (e.g., Miller et al., 2014; Thompson et al., 2017), which do not show any large winter fraction (~10%) of CH$_4$ emissions in the HBL, compared to our results for East (22 %, see Sect. 3.5.2) in the cold season.

*24. Sections 3.5.1 and 3.5.2 seem to focus more on flux totals than on the spatial distribution of fluxes (which is the title of Sect. 3.5). I would consider renaming Sect. 3.5 accordingly.*

We have renamed Sect. 3.5 from 'Spatial distribution of the fluxes' to 'National and regional distribution of annual fluxes.'

*25. Lines 646 - 662: These lines seem like they might fit better in Sect. 3.6 than in Sect. 3.5.*

We have moved these lines to Sect. 3.6 Winter natural CH$_4$ emissions. And the previous "Sect. 3.6" is now the subsection, 3.6.1. We hope this way follows better:

**3.6 Winter natural CH$_4$ emissions**

Results for cold season natural CH$_4$ fluxes are wide ranging among recent studies, as cold season natural CH$_4$ fluxes are difficult to measure and quite variable in wetland model estimates. Treat et al (2018) reported measured cold (non-growing) season fraction of wetland CH$_4$, 16 % (95 % confidence interval CI, 11.0–23.0 %) between 40˚ N and 60˚ N, and 17 % (CI 16.0–23.3 %) for north of 60˚ N. These fractions tend to be higher than process-based models (4–17 % within 40–60˚ N), while the upscaled flux estimates based on the flux measurement with machine learning technique (Peltola et al., 2019) showed cold season emission (November to March) ~20 % for north of 45˚ N. Pelletier et al. (2007) reported up to 13 % of the annual emission in the winter (November to March), in peatland in James Bay Lowland, along the Hudson Bay coastline in Canada. A recently published CH$_4$ flux dataset from the flux measurement global network (FLUXNET-CH$_4$) has a considerable contribution of cold months (October to March) to annual CH$_4$ flux, 18.1 ± 3.6 % and 15.3 ± 0.1 % in northern (> 60˚ N) and temperate (40°–60˚ N) regions, respectably (Delwiche et al., 2021). An inter-comparison of 16 wetland models from the Global Carbon Project (Ito et al., 2023) showed cold season CH4 fluxes (September to May) ranging from 11.6–40.1% in the Arctic (> 60˚ N), and 21.6–54 % north of 45˚ N. For comparison, our cold season (September to May) natural CH$_4$ emissions are 38.5 (38–39) % in the Arctic (> 60˚ N) and 51 (49–52) % north of 45˚ N. The natural CH$_4$ emission in this study is not directly comparable to the other

wetland emissions as our natural $CH_4$ emission is limited to the model domain of Canada and includes biomass burning, and soil sink. But our natural $CH_4$ emission estimate appears to be within the range of results of other studies. As the range of possible winter wetland emission fraction is large in previous studies, evidence of winter wetland/natural $CH_4$ emissions in our atmospheric $CH_4$ measurements is further examined in the following section.

**3.6.1 Signals of winter natural $CH_4$ emissions in observations**

26. *Section 3.6: What do you think is the overall take-away message or main scientific result of this section? I think it could be helpful to emphasize the main takeaway messages here. The text at the end of the section states that the diurnal cycle is consistent with the inverse modeling results, but I'm not sure if there are other key take-aways in this section. I'm also not sure to what extent the results in this section really validate the inverse model; they're certainly not inconsistent with the inverse model, but I don't know that they specifically validate or provide evaluation of the inverse model.*

As we mentioned at the end of the first paragraph in this section, examining the diurnal cycle provides observational evidence of the existence of winter natural $CH_4$ emissions. We believe this is a novel approach to detect the winter emissions using atmospheric measurements, although it does not provide quantitative verification to inverse model. It could be considered as supporting evidence that winter natural $CH_4$ emissions are present. Before making more definite statements, we need to explore if such diurnal cycle $CH_4$ signals could be used to quantify flux rates, possibly using Radon ratios such as Vogel et al. (2012). Winter flux measurements in boreal regions are still sparse because the fluxes are too weak to detect, and the field conditions often are too hush. Beyond a site-level flux measurement (though eddy covariance or chamber), the resulting evidence also indicates that the winter $CH_4$ emissions occur on a large spatial scale.

27. *Conclusions: Several topic sentences in the conclusions sentences start out with relatively technical references to case "Inv_4R12S." Instead, I would try to focus on the high-level take-away messages and avoid very technical abbreviations in this section.*

We have replaced 'Inv_4R12S' from this section with 'reference inversion.'

---

## Author Comment (AC2)

**Reply to Comments by Referee #2**

We thank the referee for the time spent on commenting on the paper, and for the encouraging comments and useful suggestions which have helped improve the paper. Below are our responses; the referee's comments are copied *in italic and red*. In the responses, we also indicate the changes made in the manuscript in blue.

*The paper deals with the estimation of Canadian methane emissions for the period 2007-2017. It is a valuable contribution to the topic, performing an ensemble regional inversion constrained with the Environment and Climate Change Canada (ECCC) surface measurement network and provides a new perspective on the important region of Canada, where large uncertainties in the methane budget have previously been recognised. The modelling methodology is well established and the ECCC data are widely used and of high quality. A couple of questions remain to be clarified:*

1. *l 15: what are the gradients mentioned here?*

By "gradients", we mean "the difference of atmospheric CH4 from one site to another". Line 15 has been changed as follows:

First, the modelled differences in atmospheric $CH_4$ among the sites show improvement after inversion when compared to observations, implying the $CH_4$ observation differences could help verify the inversion results.

2. *l 50: could maybe mention that some studies indicate a decreasing methane emission trend in the future due to increased evapotranspiration and drying of the soil (Kwon et al, 2022, https://onlinelibrary.wiley.com/doi/10.1111/gcb.16394)*

Thank you for your suggestion. We included the reference, as below:

Recent studies on the arctic and boreal peatlands reveal more complex sensitivities of $CH_4$ flux exchange. Kwon et al. (2022) noted a decreasing methane emission trend in the future due to increasing evapotranspiration and drying of the soil.

3. *105: LacLaBiche, Egbert and Downsview still have high variability after data selection. I wonder how you ensure that this is not very local influence, as, if I am not wrong, Egbert and Downsview appear to be within an urban area. Did you use any additional selection methods for these sites, e.g. wind speed?*

We did not use any additional selection to these three sites, Lac La Biche, Egbert and Downsview. At the data selection with the curve fitting technique, we applied the same selection criteria to all the sites. Figure 2 shows the hourly data as well as the afternoon mean values of atmospheric $CH_4$ mixing ratios at the individual sites. Plotting together with the hourly data makes the three sites, highly variable. As discussed in Sect. 3.6, the influence of local-scale emission strength primarily manifests in the diurnal variations of atmospheric mixing ratios. In general, afternoon mean mixing ratios represent the source strengths over a large-scale area, exhibiting synoptic-scale events in atmospheric $CH_4$. However, even during the daytime, local (point) sources can interact with local-scale atmospheric condition, leading abrupt or sporadic spikes in observed atmospheric $CH_4$. Most of these observations of high mixing ratios are removed as outliers through this selection.

4. *167: Five days is quite short time for calculating backward trajectories. See e.g. Wittig et al.: The 10-d transport backwards in time in FLEXPART is much smaller than the average residence time of air masses (typically few weeks) in the Arctic. Therefore, part of the influence of Arctic fluxes on observations can be diluted in the background. On the other hand, backward simulations over several weeks would require a very large number of particles to be accurate, at the expense of very high computational costs. Thus, I would suggest doing a test with at least 10-d trajectories.*

The duration of the backward trajectories depends on the spatial domain of interest. Wittig et al. (2023) and our study have distinctive differences in scope and inversion setting.

We focused on Canada's $CH_4$ flux estimation, using the synoptic-scale variations of daily observations. The majority of footprint information, >80%, used to optimize Canada's $CH_4$ fluxes comes from the first three days on the course of the back trajectory (Figure R1a, in our responses to the referee #1). Wittig et al. (2023) used monthly observational data to estimate the $CH_4$ budget in the Arctic countries, including Canada. Their Figure 8 illustrates that less than around 10 % of the observational information is used to estimate the fluxes, while 60-75% to constrain the background atmospheric $CH_4$ and the rest is lost in noise.

Figure 7 in Wittig et al. (2023) shows that their inversion is most sensitive to the observations in coastal sites in and around the Arctic Circle, including Alert, Canada. On the other hand, we did not use Alert because it is far north of the continental land sources and thus less influenced by the $CH_4$ fluxes in Canada.

We conducted experiments to see how a 10-day footprint would affect the flux estimations. The choice of the back trajectory duration slightly changed the modelled prior mixing ratio, which was comparable to ~10% of the uncertainty with the transport error from the different transport models used in this study (see Fig. R1b in our responses to Referee #1). The difference in the estimated fluxes between 5- and 10-day back trajectories is not significant, less than 5 % for regional fluxes. Thus, we concluded that 5-day back trajectories are sufficient in the scope of our study.

5.  205: It would be helpful to see the time series or all priors, to ensure that there are no inconsistencies during the time period studied (as there might be in EDGAR time series) that might affect the trend calculation.

For anthropogenic, ECCC-AQ2013 (ECAQ) is monthly and non-varying interannually, which was repeatedly used for the entire study period. EDGAR v4.3.2 (EDGAR) covers up to 2012. After 2012, we used the same emission field for 2012 (Fig. R2). There is a slight downward trend in EDGAR in the early period, but the change is relatively small compared to the interannually varying Wetland $CH_4$ emissions, such as WetCHARTs and CLASSIC (also see Figure S3).

[Figure]

**Figure R2. Time-series of prior annual $CH_4$ emissions. anthropogenic sources (left) and together with wetland sources (right).**

As seen in Figs S5 and S6, the choice of anthropogenic emissions results in differences of the absolute values of the estimated $CH_4$ emission in Western Canada, but not the trends.   Such differences are smaller than the spread/uncertainty of the posterior fluxes. The difference of posterior emissions in West seems to be related to the difference in the spatial distribution of the two prior anthropogenic emissions (see Fig. 3b).

We have added the time serieses of prior means to Fig. 7 as below.  It would help to see there are no apparent correlation with the trend between the priors and the posterior fluxes.

[Figure]

**Figure 7. Trend of estimated yearly CH₄ fluxes in Canada and western (West_2) and eastern (East_2) subregions from three inversion setups, 72 experiments in total. Lines show mean fluxes over each of three inversion sets with different subregion masks and observation site selections. The shaded areas indicate the range of maximum and minimum estimates among 24 experiments per inversion setup. Black dotted lines indicate mean prior emissions**.

6. *217: How do you re-grid the coarser resolution data on e.g wetland extent for use in higher resolution inversions?*

CLASSIC wetland CH₄ flux data are on a grid of 2.81°, which is only the coarser resolution than 1°×1°. First, we divided the coarse grids into small parts of 0.01° × 0.01° and then sum-up the small tiles over a grid of 1°×1° (see example below).

[Figure]

**Figure R3. Example of re-griding of CLASSIC wetland CH₄ data. Original emission map at a resolution of 2.81° (left) and re-gridded 1°×1° emission map**.

7. *273: More recent EDGAR releases include an annual cycle for the anthropogenic emissions. How would this affect your results?*

The impact of the annual cycle of anthropogenic emissions on the flux estimates is negligible in this study.

We did not use the seasonal varying EDGAR emissions. Instead, ECAQ includes larger seasonal variations than the seasonal EDGAR, especially in CH₄ emissions from Landfill sector. However, the seasonal variation of ECAQ anthropogenic emission is less visible, compared to the one in wetland CH₄ emissions as Figs. 3e and 3f in the manuscript. In our posterior fluxes, no clear seasonality due to the seasonally varying ECAQ was found.

8. *291: In reality, cold months can vary from year to another and the shoulder seasons may have a significant impact on methane emissions. How would this affect your results?*

There are year-to-year changes in the seasonal variations of posterior fluxes. In this study, we analyse the mean seasonal cycle over the study period. In this way, inter-annual variations are minimized in our analysis.

9. *Figure6: What could be the midwinter peak in methane emissions (especially 2011 and 2013 in East)? Anthropogenic or natural emissions?*

Figure R4 shows the year-to-year variations in the January footprints for Fraserdale. In 2011 and 2013, there appears to be more footprint influence from the province of Manitoba (with anthropogenic emissions from cities like Winnipeg) on the western end of the sub-region East. Thus, the notable winter methane emission peaks in East for 2011 and 2013 could be a combination of natural and anthropogenic emissions. In this study, the flux partitioning into natural and anthropogenic is based on the mean seasonal $CH_4$ fluxes. More analysis is needed to understand the inter-annual variations in the flux partitioning in the site observations.

[Figure]

**Figure R4. Monthly mean footprints in January 2011 to 2013 (left to right) for Fraserdale (FSD). Unit is $log_{10}$ ppm/(mol /($m^2$ s)).**

10. *Table S2: Can you give a statistical estimate of the significance of the trend, in addition to SD among the ensembles?*

We have added p-value of the trend in Table S2 as follows:

**Table S2.** Ensemble mean trends of estimated yearly $CH_4$ fluxes,  uncertainties (SD among the ensembles) and *p*-values for Canada and western (West_2) and eastern (East_2) regions from three inversion setups, 72 experiments in total and 24 experiments in each of three inversion setups (Inv_4R12S, Inv_2R12S and Inv_2R2S) with different subregion masks and observation site selections. The trends are calculated as slopes of linear fit over three periods: the whole (2007–2017), the early (2007–2011) and the later (2012–2017) periods.

| | Canada | | | West_2 | | | East_2 | | |
|---|---|---|---|---|---|---|---|---|---|
| | 2007–2017 | 2007–2011 | 2012–2017 | 2007–2017 | 2007–2011 | 2012–2017 | 2007–2017 | 2007–2011 | 2012–2017 |
| **Total** | -0.20 ± 0.14 $p = 0.08$ | -0.36 ± 0.59 $p = 0.09$ | 0.05 ± 0.23 $p = 0.64$ | -0.15 ±0.14 $p = 0.53$ | -0.49 ± 0.48 $p = 0.23$ | 0.02 ± 0.17 $p = 0.64$ | -0.05 ± 0.10 $p = 0.77$ | 0.02 ± 0.27 $p = 0.92$ | 0.03 ± 0.12 $p = 0.65$ |
| **Inv_4R12S** | -0.14 ± 0.18 $p = 0.13$ | -0.42 ± 0.68 $p = 0.13$ | 0.00 ± 0.19 $p = 0.02$ | -0.12 ±0.14 $p = 0.34$ | -0.59 ± 0.53 $p = 0.07$ | -0.07 ± 0.14 $p = 0.46$ | -0.02 ± 0.06 $p = 0.98$ | 0.16 ± 0.22 $p = 0.61$ | -0.02 ± 0.04 $p = 0.48$ |
| **Inv_2R12S** | -0.25 ± 0.13 $p = 0.03$ | -0.44 ± 0.62 $p = 0.01$ | -0.04 ± 0.22 $p = 0.53$ | -0.27 ±0.13 $p = 0.40$ | -0.70 ± 0.47 $p = 0.11$ | -0.01 ± 0.15 $p = 0.24$ | 0.03 ± 0.06 $p = 0.79$ | 0.25 ± 0.22 $p = 0.95$ | -0.03 ± 0.10 $p = 0.51$ |
| **Inv_2R2S** | -0.22 ± 0.08 $p = 0.17$ | -0.22 ± 0.43 $p = 0.52$ | 0.18 ± 0.23 $p = 0.10$ | -0.07 ±0.05 $p = 0.77$ | -0.18 ± 0.26 $p = 0.72$ | 0.12 ± 0.16 $p = 0.91$ | -0.15 ± 0.08 $p = 0.47$ | -0.04 ± 0.29 $p = 0..63$ | 0.06 ± 0.11 $p = 0.58$ |

[Figure]

-0.5  -0.3  -0.1  0.1  0.3  0.5   (Tg CH$_4$ year$^{-2}$)

11. *396: Could you use trajectories to select those time periods when air masses were transported directly from one site to the other?*

We could not make such selection. The wind direction is always changeable. The air is diffusive over time. Because of these characteristics of the air, the likelihood of an air mass transported directly from one site to another is low. Therefore, we compared the multi-year mean concentration difference of the two sites (representing the mean transport) by month, without any selection based on wind direction/trajectories.

12. *465: How about the increase in the depth of the permafrost thaw layer, which progresses through the summer and increases the temperature of the subsurface layers? Could it, in part, explain the later emission maximum?*

It could be, but not sure. In the Arctic, the seasonally thawed active permafrost layer increases in depth during the summer and starts freezing in the late August or September. The thawed permafrost increases the soil moisture, providing a favourable environment for the microbial CH$_4$ production. In the mid-latitude subregions, West and East in this study, there are some small areas of discontinuous permafrost layers, and sporadic or isolated permafrost patches in the deep soil (e.g., Tarnocai, et al. 2009). This type of permafrost may not be actively impact on the seasonality of Wetland CH$_4$ emissions.

*13. 694: Could it be possible that the diurnal cycle may also be influenced by anthropogenic sources, as the LLC was described as having oil industry in the south of the site.*

From this context, we assume that LLC is meant to be LLB (Lac La Biche). ECCC has another monitoring station near LLB (~400 –500 km away) at Esther (EST) in CHOPS (Cold Heavy Oil Production with Sand) region with known local anthropogenic sources. Figure R5 show the seasonal cycles of normalized diurnal amplitude at EST and SD along with LLB. On average, winter diurnal amplitudes at both sites are similar. However, the amplitude at EST is lower than SD. This result indicates that the strength and spatial distribution of local $CH_4$ sources around EST are irregular in time as the local anthropogenic $CH_4$ sources are fugitive from the oil/gas industry. In contrast to EST, the diurnal amplitude at LLB is more prominent than SD around the year. This distinctive difference between LLB and EST supports the existence of regular $CH_4$ sources around the LLB; that is, they are presumably natural sources.

[Figure]

**Figure R5. Seasonal cycle of normalized diurnal amplitude and SD of observed atmospheric $CH_4$ during the afternoon mean (14–16 local time based of normalization) for Lac La Biche (LLB, same as shown in Fig 14), and Esther (EST).**

**References**

Kwon, M. J., Ballantyne, A., Ciais, P., Qiu, C., Salmon, E., Raoult, N., Guenet, B., Göckede, M., Euskirchen, E. S., Nykänen, H., Schuur, E. A. G., Turetsky, M. R., Dieleman, C. M., Kane, E. S., and Zona, D.: Lowering water table reduces carbon sink strength and carbon stocks in northern peatlands, Global Change Biol., 28, 6752-6770, https://doi.org/10.1111/gcb.16394, 2022.

Tarnocai, C., Canadell, J. G., Schuur, E. A. G., Kuhry, P., Mazhitova, G., and Zimov, S.: Soil organic carbon pools in the northern circumpolar permafrost region, Global Biogeochem. Cycles, 23, https://doi.org/10.1029/2008GB003327, 2009.

Wittig, S., Berchet, A., Pison, I., Saunois, M., Thanwerdas, J., Martinez, A., Paris, J. D., Machida, T., Sasakawa, M., Worthy, D. E. J., Lan, X., Thompson, R. L., Sollum, E., and Arshinov, M.: Estimating methane emissions in the Arctic nations using surface observations from 2008 to 2019, Atmos. Chem. Phys., 23, 6457-6485, https://doi.org/10.5194/acp-23-6457-2023, 2023.